# The mTOR pathway is necessary for survival of mice with short telomeres

Iole Ferrara-Romeo[1], Paula Martinez[1], Sarita Saraswati[1], Kurt Whittemore[1], Osvaldo Graña-Castro[2], Lydia Thelma Poluha[1], Rosa Serrano[1], Elena Hernandez-Encinas[3], Carmen Blanco-Aparicio [3], Juana Maria Flores[4] & Maria A. Blasco[1✉]

Telomerase deficiency leads to age-related diseases and shorter lifespans. Inhibition of the mechanistic target of rapamycin (mTOR) delays aging and age-related pathologies. Here, we show that telomerase deficient mice with short telomeres (G2-$Terc^{-/-}$) have an hyper-activated mTOR pathway with increased levels of phosphorylated ribosomal S6 protein in liver, skeletal muscle and heart, a target of mTORC1. Transcriptional profiling confirms mTOR activation in G2-$Terc^{-/-}$ livers. Treatment of G2-$Terc^{-/-}$ mice with rapamycin, an inhibitor of mTORC1, decreases survival, in contrast to lifespan extension in wild-type controls. Deletion of mTORC1 downstream S6 kinase 1 in G3-$Terc^{-/-}$ mice also decreases longevity, in contrast to lifespan extension in single $S6K1^{-/-}$ female mice. These findings demonstrate that mTOR is important for survival in the context of short telomeres, and that its inhibition is deleterious in this setting. These results are of clinical interest in the case of human syndromes characterized by critically short telomeres.

[1] Telomeres and Telomerase Group, Molecular Oncology Program, Spanish National Cancer Centre (CNIO), Melchor Fernández Almagro 3, E-28029 Madrid, Spain. [2] Bioinformatics Unit, Structural Biology and Biocomputing Program, Spanish National Cancer Centre (CNIO), Melchor Fernández Almagro 3, E-28029 Madrid, Spain. [3] Experimental Therapeutics Program, Spanish National Cancer Centre (CNIO), Melchor Fernández Almagro 3, E-28029 Madrid, Spain. [4] Animal Surgery and Medicine Department, Faculty of Veterinary Science, Complutense University of Madrid, Avenida Puerta de Hierro s/n, E-28040 Madrid, Spain. ✉email: mblasco@cnio.es

Telomeres are nucleoprotein structures that protect the ends of chromosomes from being recognized as DNA breaks[1,2]. Mammalian telomeres are composed of repeats of the TTAGGG DNA sequence bound by a six-protein complex termed shelterin[2]. Owing to the end-replication problem[3] telomeres shorten with each cell division leading to progressive telomere attrition, which is considered one of the mechanisms underlying organismal ageing[4,5]. When telomeres become critically short, they trigger a persistent DNA damage response (DDR) at the chromosome ends[6], chromosomal end-to-end fusions as well as cellular senescence and/or apoptosis, eventually compromising the regenerative capacity of tissues[7]. Telomerase is a reverse transcriptase capable of synthesizing telomeric repeats de novo, thus elongating telomeres[8]. Telomerase is composed of a catalytic subunit (TERT) and an associated RNA component (Terc), which serves as a template for the synthesis of TTAGGG repeats[8]. Telomerase is inactive in adult tissues with the exception of some stem cells compartments; however, this is not sufficient to prevent telomere attrition with age in tissues[4,7,9]. In contrast, cancer cells aberrantly reactivate telomerase to maintain telomeres and divide indefinitely[10]. Indeed, TERT is one of the most mutated genes in human cancers[11].

Mice genetically deficient for the RNA component of telomerase ($Terc^{-/-}$) show accelerated telomere shortening and decreased lifespan owing to premature development of age-associated pathologies, being intestinal atrophy the most prevalent[12–14]. These pathologies are anticipated with increasing generations of telomerase-deficient mice owing to inheritance of progressively shorter telomeres with each mouse generation[12–14]. Owing to the fact that telomere maintenance by telomerase is essential for tumor growth, telomerase-deficient mice with short telomeres are cancer resistant, except when in the absence of p53, a potent inducer of cell cycle arrest and/or apoptosis in response to telomere DNA damage[15,16].

There are a number of human diseases, known as telomere syndromes, that are characterized by the presence of abnormally short telomeres caused by mutations in telomerase and other telomere genes[17,18]. These diseases include cases of Dyskeratosis congenita, aplastic anemia, as well as pulmonary and liver fibrosis among other degenerative diseases. Unfortunately, there are no effective therapeutic strategies for the treatment of these diseases in the clinic.

Inhibition of the nutrient sensing pathway regulated by the mechanistic target of rapamycin (mTOR) is considered a therapeutic target to delay aging and age-related pathologies. mTOR is a serine/threonin protein kinase of the PI3K-related family that is part of the PI3K/AKT signaling pathway, and that regulated cell growth and metabolism in response to nutrient availability and hormonal cues[19,20]. mTOR exists in two distinct complexes, mTORC1 and mTORC2, each with different substrates and activities[21,22]. mTORC1 downstream targets include ribosomal protein S6 kinase (S6K), eukaryotic initiation factor 4E binding protein-1 (4EBP1), and unc-51 like kinase (ULK1)[21,22]. mTORC2 downstream substrates include the protein kinase AKT, serum- and glucocorticoid-induced kinase (SGK), and protein kinase C (PKCα)[21,22]. Of these two complexes, mTORC1 is the only one sensitive to acute rapamycin treatment[19,20,23]. However, prolonged rapamycin treatment has also been shown to inhibit mTORC2 activity[24,25].

Genetic or pharmacological inhibition of mTORC1 with rapamycin, or with rapamycin-derived compounds known as rapalogs, delays aging and can increase the lifespan of yeast, flies, worms, and mice[26–30]. In mice, rapamycin treatment can increase longevity even in the case of outbreed mouse strains[29–31]. Similarly, deletion of the ribosomal S6 protein kinase 1 (S6K1), a downstream effector of mTORC1, can also increase lifespan in female mice[32]. Furthermore, evidence suggests that lifespan extension by dietary restriction may also be partly due to inhibition of the mTORC1 function[33,34]. Rapamycin has also been shown to significantly decrease cancer incidence in wild-type mice as well as to have immunosuppressant properties[35,36].

In light of all these beneficial effects of inhibition of the mTORC1 pathway in extending longevity, here we set to address whether rapamycin treatment could also ameliorate the premature aging phenotypes and the decreased longevity of telomerase-deficient mice with short telomeres. This is of relevance as mTOR inhibitors could represent potential treatments for human patients suffering from telomere syndromes.

Here we find that the mTOR pathway is upregulated in telomerase-deficient mice with short telomeres. Unexpectedly, inhibition of the mTOR pathway both by chronic rapamycin treatment and by genetic means decreases longevity of $Terc^{-/-}$ mice, in marked contrast to lifespan extension in similarly treated wild-type mice. Together, these findings demonstrate that hyperactivation of the mTOR pathway in the context of telomerase deficiency and short telomeres is acting as a survival pathway, and that inhibition of this pathway has deleterious effects in this condition.

## Results

**Chronic rapamycin diet decreases lifespan of $Terc^{-/-}$ mice.** To address whether rapamycin treatment could ameliorate premature aging pathologies and decreased longevity in mice with short telomeres, 3-month-old wild type and second-generation telomerase-deficient mice (G2 $Terc^{-/-}$) in a C57BL/6 genetic background[14] were fed with either control chow or chow-containing encapsulated rapamycin at 42 ppm (mg of drug per kg of food)[31]. Mouse cohorts were followed until the humanitarian endpoint to determine both median and maximum longevity (Fig. 1a). We found that rapamycin treatment increased median longevity of wild-type mice ($Terc^{+/+}$) by 39%, resulting in a median longevity of 26.5 months in rapamycin-fed $Terc^{+/+}$ mice compared to only 19 months in the control-fed cohorts (Fig. 1b). This was increased to 58% when considering tumor-free survival (Fig. 1c). In addition, maximum lifespan (mean lifespan of the 10% oldest individuals within each cohort) was also significantly increased, reaching 32 months in the case of rapamycin-fed $Terc^{+/+}$ mice compared to 29.25 months in control diet-fed $Terc^{+/+}$ mice (Fig. 1b). When survival curves were separated by sex, rapamycin-fed $Terc^{+/+}$ females showed an increase in median lifespan of 23% compared to control diet-fed females, while the increase was of 43% in the case of the rapamycin-fed males compared to control-diet $Terc^{+/+}$ males (Supplementary Fig. 1A, B). We observed similar increases in longevity when considering tumor-free survival (Supplementary Fig. 1C, D).

In contrast to the observed lifespan extension in rapamycin-fed wild-type mice, rapamycin-fed telomerase-deficient mice showed the opposite effect, with a significantly decreased longevity upon rapamycin treatment (Fig. 1b, c). In particular, rapamycin-fed G2 $Terc^{-/-}$ mice showed a 16% decrease in median lifespan compared to control fed G2 $Terc^{-/-}$ mice (Fig. 1b). When G2 $Terc^{-/-}$ mice were separated by sex, median survival was decreased by 19% in the rapamycin-fed G2 $Terc^{-/-}$ males compared to control-fed males, while no changes in median survival were observed between the rapamycin-fed G2 $Terc^{-/-}$ females and the G2 $Terc^{-/-}$ controls (Supplementary Fig. 1A, B). We obtained similar results when considering tumor-free survival (Supplementary Fig. 1C, D). These findings suggest that lifespan extension by rapamycin is abrogated in the context of telomerase deficiency and presence of short telomeres.

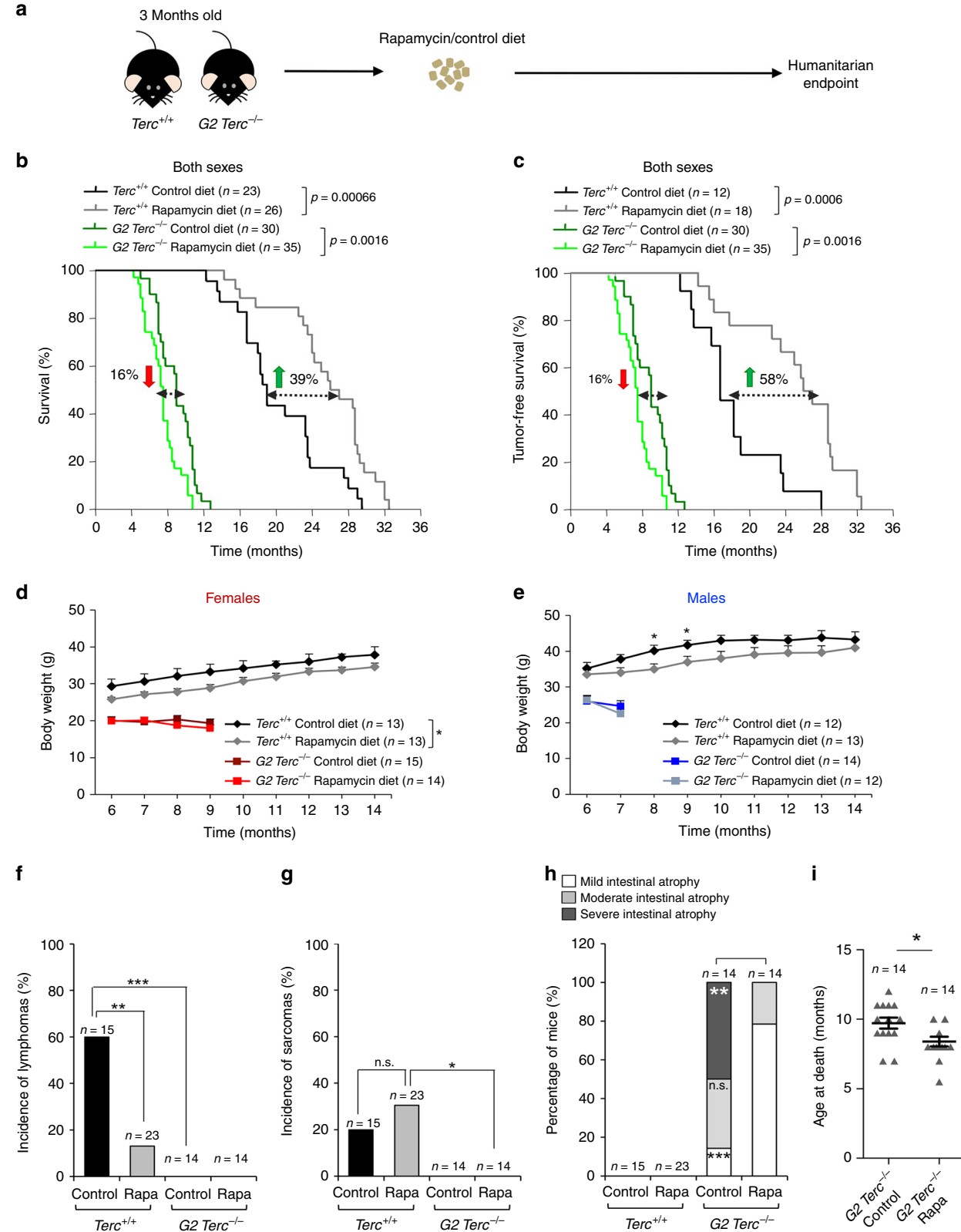

One of the main phenotypes of chronic rapamycin treatment in mice is a significant decrease in body weight owing to the known effects of rapamycin on metabolism[30]. In agreement with this, rapamycin-fed $Terc^{+/+}$ male and female mice showed a decrease in body weight compared to control diet-fed counterparts (Fig. 1d, e). G2 $Terc^{-/-}$ mice of both sexes started off with

smaller body weights compared to wild-type mice (Fig. 1d, e)[14]. Rapamycin treatment did not further decrease body weight of G2 $Terc^{-/-}$ mice, suggesting that this phenotype associated to rapamycin was abolished in G2 $Terc^{-/-}$ mice (Fig. 1d, e). To study whether G2 $Terc^{-/-}$ mice had upregulated the xenobiotic response pathway resulting in degradation of the rapamycin in

**Fig. 1 Chronic rapamycin treatment decreases the lifespan of $Terc^{-/-}$ mice. a** Three-month-old $Terc^{+/+}$ and G2 $Terc^{-/-}$ mice were fed control chow or chow-containing encapsulated rapamycin at 42 ppm and followed until the humanitarian endpoint (HEP). **b, c** Kaplan–Meier survival curves of $Terc^{+/+}$ and G2 $Terc^{-/-}$ mice of both sexes fed rapamycin or control diet **b**. Kaplan–Meier tumor-free survival curves, including only mice that did not present any neoplastic pathology at the time of death (**c**). The variation of rapamycin-fed mice median survival is indicated as the percentage of that of the control-fed mice of the same genotype; green arrows: rapamycin-mediated increase in median survival; red arrows: rapamycin-mediated decrease in median survival. Statistical significance was determined by the log-rank test. The $p$ values are indicated. **d, e** Body weight changes in female (**d**) and male (**e**) mice of both genotypes fed rapamycin or control diet. Statistical significance was determined by two-tailed Student's $t$-test. **f, g** Incidence of lymphomas (**f**) and sarcomas (**g**) in the four groups of mice. **h** The percentage of mice presenting mild, medium, or severe intestinal atrophy according to histopathological analysis. For a detailed histological description see Methods. A chi-square test was used to calculate statistical differences in the incidence of both tumors and intestinal lesions. **i** Mean age at the HEP of the G2 $Terc^{-/-}$ mice used for the histopathological analysis in **h**. A two-tailed Student's $t$-test was used to calculate the statistical significance. Error bars represent the standard error (SE). $n$ = number of mice; $*p \le 0.05$; $**p \le 0.01$; $***p \le 0.001$; n.s. not significant. Source data are provided as a Source Data file.

the liver and thereby blocking its effects on survival, we measured the rapamycin levels in fed male and female liver samples as well as in fasted and fed male plasma samples (Supplementary Fig. 2A, B). We found similar liver rapamycin levels in $Terc^{+/+}$ and G2 $Terc^{-/-}$ males and females (Supplementary Fig. 2A). We also detected similar rapamycin plasma levels in $Terc^{+/+}$ and G2 $Terc^{-/-}$ samples in both nutritional conditions, fasted and fed (Supplementary Fig. 2B). These observations rule out a telomerase-dependent degradation of rapamycin.

**Cancer and aging pathologies in rapamycin-fed $Terc^{-/-}$ mice.** To further investigate the higher mortality of rapamycin-fed G2 $Terc^{-/-}$ mice, we performed a full histopathological analysis at death point in all mouse cohorts. As expected[37], rapamycin-fed wild-type mice showed significantly decreased lymphoma incidence (Fig. 1f), although the incidence of sarcoma was not affected by rapamycin (Fig. 1g). $Terc^{-/-}$ mice are reported to be cancer resistant owing to a tumor suppressive role of short telomeres, with the exception of p53-deficiency[16]. In agreement with this, control diet-fed G2 $Terc^{-/-}$ mice did not show tumors at their time of death and this was not modified by rapamycin treatment (Fig. 1f, g), thus ruling out that the decreased survival of rapamycin-treated G2 $Terc^{-/-}$ mice was due to loss of the tumor suppressor effect of short telomeres[15,16]. In agreement with this, the tumor-free survival curves of wild type and G2 $Terc^{-/-}$ mice showed the same trends than the overall survival curves (Fig. 1b, c), also when separated by sex (Supplementary Fig. 1C,D).

We next addressed whether rapamycin treatment could be aggravating one of the main causes of death of G2 $Terc^{-/-}$ mice, i.e., intestinal atrophy[14]. To this end, we classified the intestinal pathologies present at the mouse endpoint in mild, medium, or severe according to the pathological findings. Mild intestinal lesions are characterized by multifocal epithelia and glandular atrophy that affect up to 20% of the tissue. Medium and severe intestinal lesions show multifocal areas in the mucosa lacking glands and the presence of degenerative epithelial and glandular cystic hyperplasia that affect between 21% and 60% or >61% of the tissue, respectively. We found that 100% of G2 $Terc^{-/-}$ mice presented intestinal atrophy at their endpoint in contrast to the absence of this pathology in wild-type cohorts independently of the diet (Fig. 1h). Rapamycin-fed G2 $Terc^{-/-}$ mice showed less severe intestinal atrophies compared to the control diet cohorts (Fig. 1h). In particular, while 50% of control diet G2 $Terc^{-/-}$ mice presented severe intestinal atrophy, 80% of rapamycin-fed G2 $Terc^{-/-}$ mice showed mild atrophy (Fig. 1h). Although these findings may suggest that rapamycin treatment ameliorates intestinal atrophy in telomerase-deficient mice, the fact that rapamycin-treated G2 $Terc^{-/-}$ mice died at an earlier timepoint (2 months earlier) compared to the control diet cohorts (Fig. 1i)

may also explain the lower severity of intestinal atrophy. This notion is supported by the fact that, when we separated mice by sex, only rapamycin-treated G2 $Terc^{-/-}$ males but not females showed significantly decreased severe intestinal lesions (Supplementary Fig. 2C, D), in agreement with the fact that only rapamycin-treated males showed a significantly decreased longevity compared to G2 $Terc^{-/-}$ controls (see Supplementary Figs. 1B and 2C, D).

Together, these findings show that the negative effects of rapamycin treatment on telomerase-deficient mice are not due to increased cancer or to an aggravation of the degenerative pathologies associated to telomerase deficiency and the presence of short telomeres.

**Rapamycin treatment does not affect telomere length.** The finding that lifespan extension by rapamycin does not occur in telomerase-deficient mice may suggest that the underlying mechanism of lifespan extension by rapamycin may require of telomere maintenance by telomerase, a possibility that has not been addressed before. To address this, we measured telomere length in $Terc^{+/+}$ and G2 $Terc^{-/-}$ female cohorts subjected either to control or rapamycin diet by performing quantitative telomere fluorescence in situ hybridization (Q-FISH) analysis in mice at the human endpoint. To avoid age-related variations in telomere length within each experimental group, we compared untreated and rapamycin-treated wild type and G2 $Terc^{-/-}$ female mice that died at same age, namely 2 years old in the case of wild-type mice and 6 months old in the case of $Terc^{-/-}$ mice. Telomere length was determined in tissue sections from a highly proliferative tissue such as the intestine and a non-proliferative post-mitotic tissue such as the liver. As expected, G2 $Terc^{-/-}$ mice showed significantly shorter telomeres than the $Terc^{+/+}$ controls in both tissues. Rapamycin treatment of wild type and G2 $Terc^{-/-}$ mice did not result in significant changes in telomere length in both tissues compared to the control diet-fed cohorts (Fig. 2a, b). Thus, the opposite effects on longevity of the rapamycin diet in wild type and G2 $Terc^{-/-}$ mice cannot be attributed to telomere length changes. To further rule out any potential effects of rapamycin on telomere length, we determined telomere length in a longitudinal manner in several of the individual mice included in the study. To this end, we obtained white blood cells from mice of both genotypes at different time points during treatment (i.e., at 4.5, 6–7, 12–15 and 15–20 months of age, as well as at their endpoint) and measured telomere length by high throughput Q-FISH (HT Q-FISH)[38]. As expected, $Terc^{-/-}$ mice under a control diet showed a faster rate of telomere shortening in peripheral blood than similarly fed wild-type mice (Fig. 2C). Rapamycin treatment did not alter the rate of telomere shortening in wild-type and telomerase-deficient mice with aging (Fig. 2c). Thus, the increased longevity of rapamycin-treated

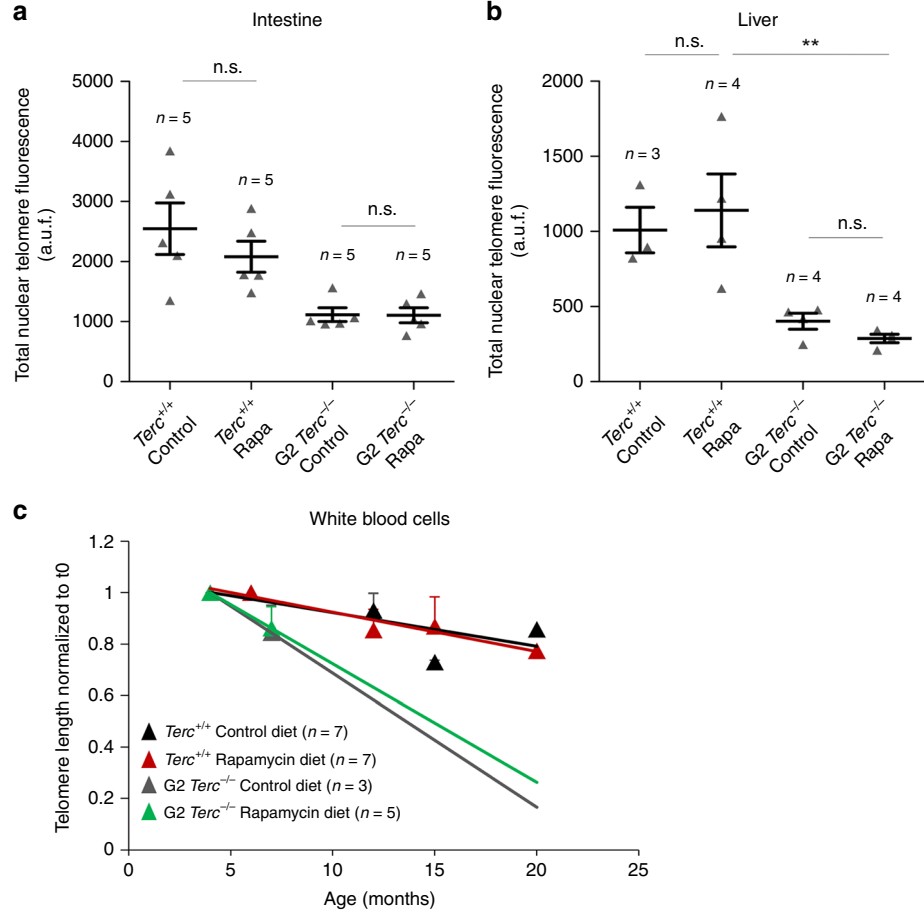

**Fig. 2 Rapamycin treatment does not influence telomere length. a**, **b** Total nuclear telomere fluorescence in intestine (**a**) and liver (**b**) sections of rapamycin- or control-fed $Terc^{+/+}$ and G2 $Terc^{-/-}$ mice at the endpoint, measured by Q-FISH. a.u.f. arbitrary units of fluorescence. Error bars represent the SE. **c** Mean telomere fluorescence in white blood cells from mice of both genotypes at 4.5, 6–7, 12–15, and 15–20 months of age, measured by high throughput Q-FISH. Values and error bars represent the mean and SE, respectively. $n =$ number of mice. Statistical significance was determined by one-way Anova with post hoc Tukey test. *$p \leq 0.05$; **$p \leq 0.01$; ***$p \leq 0.001$; n.s. not significant. Source data are provided as a Source Data file.

wild-type mice is not due to a better preservation of telomere length during aging. Similarly, rapamycin treatment did not aggravate telomere shortening in the context of the telomerase-deficient mice.

**Rapamycin treatment increases replicative damage**. Short telomeres have been previously shown to induce a persistent DDR at chromosome ends, which in turn can trigger senescence and/or apoptosis and limit the regenerative capacity of tissues[39]. To address whether rapamycin treatment increases DNA damage or exacerbates the proliferative defects in the context of telomerase deficiency, we next studied expression of the DNA damage marker γH2AX, of the senescence markers p53 and p21, and of the apoptosis marker active caspase 3 (AC3) in both $Terc^{+/+}$ and G2 $Terc^{-/-}$ intestines at the human endpoint. As expected, G2 $Terc^{-/-}$ mice showed more cells positive for these markers in the intestine compared to the wild-type controls (Supplementary Fig. 3A–D). The number of telomere-induced DNA damage foci (TIFs), indicative of telomere damage, was also higher in G2 $Terc^{-/-}$ intestines compared to wild-type mice although the difference did not reach statistical significance (Supplementary Fig. 3E). Rapamycin treatment did not alter the number of cells positive for these markers in the intestines of wild type or G2 $Terc^{-/-}$ mice (Supplementary Fig. 3A–E). We also analyzed the levels of γH2AX, p53, p19, and the mitotic marker pH3 in both

$Terc^{+/+}$ and G2 $Terc^{-/-}$ intestines of healthy young mice subjected to rapamycin treatment for 2 months (Supplementary Fig. 3F–I). Again, we did not find significant differences as the consequence of rapamycin treatment (Supplementary Fig. 3F–I). These results were also confirmed in a non-proliferative tissue such as skeletal muscle in mice treated with rapamycin for 2 months (Supplementary Fig. 3J–L).

Critically short-uncapped telomeres can lead to chromosomal aberrations, including end-to-end chromosome fusions[12]. We next determined whether rapamycin treatment could affect the occurrence of chromosomal aberrations. For that, we treated both $Terc^{+/+}$ and G2 $Terc^{-/-}$ mouse embryonic fibroblasts (MEFs) with rapamycin (1 μM, for 24 h) and then performed telomere-FISH analysis. We observed that G2 $Terc^{-/-}$ MEFs showed significantly higher numbers of chromosome fusion events per metaphase compared to wild-type MEFs, and this was not aggravated by rapamycin treatment (Supplementary Fig. 4A, C). One type of telomere aberration are the so-called multitelomeric signals (MTS), i.e. presence of chromosome ends with more than 1 telomere signal, that are associated with telomere fragility owing to DNA replication problems at telomeres[40,41]. Rapamycin treatment of wild-type MEFs resulted in a significant increase in the MTS frequency (Supplementary Fig. 4B, C), suggestive of increased replicative damage at telomeres. Untreated G2 $Terc^{-/-}$ MEFs showed an already elevated frequency of MTSs;

however, this was not further aggravated by rapamycin treatment (Supplementary Fig. 4B, C). These findings suggest that rapamycin treatment increases replicative stress in wild-type MEFs but not in telomerase-deficient MEFs. In support of rapamycin treatment causing increased replicative damage in wild-type cells, we observed that rapamycin-treated wild-type mice showed increased numbers of cells positive for phosphorylated RPA (pRPA) in the intestine, a known marker of replicative stress[42,43] (Supplementary Fig. 4D, E). Again, telomerase-deficient mice subjected to a control diet already showed higher pRPA levels in the intestine and this was not further aggravated by rapamycin treatment (Supplementary Fig. 4D, E). Previous in vitro work with human cells showed that rapamycin does not impact on senescence mediated pathways but supress the pro-inflammatory phenotype of senescence cells (SASP) by reducing cytokine expression and IL-6 secretion[44]. We measured IL-6 plasma levels in our mouse cohorts at the human endpoint and found elevated IL-6 levels in G2 $Terc^{-/-}$ mice compared to wild-type mice, in agreement with a higher number of senescent cells in mice with short telomeres. However, rapamycin treatment did not alter IL-6 plasma levels in either genotype (Supplementary Fig. 4F). Together, these findings suggest that the lower survival of telomerase-deficient mice subjected to a rapamycin diet cannot be attributed to increased levels of global and telomeric DNA damage and increased senescence or apoptosis.

**Persistent S6 phosphorylation in $Terc^{-/-}$ mice.** As the effects of rapamycin treatment in modulating lifespan seem to be independent of telomere length and of cellular DNA damage, we set to address whether telomerase deficiency may have an impact on the regulation of the mTOR pathway per se. It is well known that the rapamycin-mediated lifespan extension in wild-type mice occurs concomitantly with the downregulation of the mTOR signaling pathway[29]. To analyze the downregulation levels of mTORC1 signaling by rapamycin, we measured the hepatic levels of the phosphorylated ribosomal protein S6 (pS6), a downstream target of mTORC1, in both $Terc^{+/+}$ and G2 $Terc^{-/-}$ mice sacrificed at the humane endpoint. As expected, rapamycin-treated $Terc^{+/+}$ livers showed significantly lower levels of pS6 than the untreated controls (Fig. 3a, b). Strikingly, pS6 levels were not decreased in rapamycin-treated G2 $Terc^{-/-}$ livers compared to the control diet cohorts (Fig. 3a, b). Thus, the ability of rapamycin to inhibit the mTOR pathway is lost or compensated in the context of telomerase-deficient mice.

As the above results were obtained at the mouse endpoint, next, we set out to analyze the mTORC1 pathway activity in young, healthy mice under similar nutritional conditions. To this end, a group of adult (2–6 months old) $Terc^{+/+}$ and G2 $Terc^{-/-}$ mice fed with rapamycin or control diet were sacrificed after 2 months of treatment (Fig. 3C). To ensure that all mice were in the same nutritional conditions at the time of sacrifice, we fasted them overnight and refed them for 3 h before sacrifice. We analyzed the pS6 levels in the liver, skeletal muscle, and heart by immunohistochemistry. We found that rapamycin-treated wild-type mice showed significant lower pS6 levels compared to control fed counterparts in the three tissues (Fig. 3D–I). Again, rapamycin-fed G2 $Terc^{-/-}$ mice did not show inhibition of pS6 levels in the liver, skeletal muscle, or in the heart compared to control-fed mice (Fig. 3d–i), suggesting that chronic rapamycin treatment does not inhibit mTORC1 in the context of $Terc^{-/-}$ mice.

Inhibition of the mTOR pathway results in glucose intolerance, insulin resistance, and downregulates glycolysis, leading to a reduction in ATP levels[25,31,45,46]. To address the rapamycin effects on the response to glucose and insulin we performed a glucose (GTT) and insulin (ITT) tolerance tests on these mouse cohorts. We found that wild-type mice treated with rapamycin are more glucose intolerant compared to untreated wild-type mice but this was not seen in G2 $Terc^{-/-}$ (Supplementary Fig. 5A–B). Rapamycin treatment however did not affect the response to exogenously administered insulin in any of the genotypes (Supplementary Fig. 5C). Plasma IGF1 levels were not altered by rapamycin in either wild type or G2 $Terc^{-/-}$ mice (Supplementary Fig. 5D).

mTOR has been shown to impact on mitochondrial biogenesis/autophagy[47]. Since $Terc^{-/-}$ mice have a decreased mitochondrial function[48], we set to address whether rapamycin treatment could be aggravating this phenotype. To this end, we analyzed the mitochondrial content and the ratio of LC3-II/LC3-I as a readout of autophagy[47] (Fig. 4). We found that mTORC1 inhibition by chronic rapamycin treatment induces autophagy in wild-type mice but does not alter autophagy levels in the G2 $Terc^{-/-}$ mice, again indicating a lack of effect of chronic rapamycin in mice with short telomeres (Fig. 4a). As expected[48], control diet-fed G2 $Terc^{-/-}$ mice presented reduced mtDNA copy number in liver and muscle compared to wild-type controls indicating worsened mitochondrial biogenesis in $Terc^{-/-}$ mice (Fig. 4b, c). Rapamycin treatment led to a decrease in the mtDNA copy number in wild-type liver and muscle samples while no effects were seen in G2 $Terc^{-/-}$ mice (Fig. 4b, c). We also analyzed by western blot the levels of ATP5A, UQCRC2, MTCO1 and SDHB, components of mitochondrial complex V, III, IV, and II, respectively, in liver samples. The results confirmed decreased mitochondria in wild type treated with rapamycin and in G2 $Terc^{-/-}$ mice compared to wild-type control samples and no differences between untreated and rapamycin-treated G2 $Terc^{-/-}$ samples (Fig. 4d). We measured the hepatic levels of ATP after 2 months of rapamycin treatment. In agreement with reduced number of mitochondria, rapamycin-treated $Terc^{+/+}$ livers showed lower levels of ATP than the untreated controls (Supplementary Fig. 5E). However, no differences in hepatic ATP levels were observed between rapamycin-treated and control-fed G2 $Terc^{-/-}$ mice (Supplementary Fig. 5E), further suggesting that inhibition of the mTOR pathway seemed to be lost or compensated in the context of telomerase-deficient mice under a chronic rapamycin treatment.

To distinguish between these two possibilities, we first tested whether G2 $Terc^{-/-}$ mice were able to inhibit mTORC1 in response to an acute rapamycin treatment. To this end, we performed an acute inhibition of the mTOR pathway by intraperitoneal injection (i.p.) of rapamycin (2 mg/kg body weight) in 4–6 months old $Terc^{+/+}$ and G2 $Terc^{-/-}$ males followed by mouse sacrifice 2 h later. We found that rapamycin was able to inhibit mTORC1 activity in both $Terc^{+/+}$ and G2 $Terc^{-/-}$ mice, as indicated by phosphorylation of its downstream target S6 in the liver by western blot (Supplementary Fig. 6). This result indicates that the lack of inhibition of mTORC1 in G2 $Terc^{-/-}$ mice under a chronic rapamycin diet is not due to a failure of rapamycin to directly inhibit mTORC1. Instead, these findings suggest that G2 $Terc^{-/-}$ mice subjected to a chronic rapamycin treatment may be compensating/over-activating the mTORC1 pathway to minimize deleterious effects of rapamycin treatment in the context of short telomeres, such as the decreased survival observed in rapamycin-treated G2 $Terc^{-/-}$ mice (Fig. 1b).

Further supporting this possibility, we observed significantly higher hepatic levels of pS6 in G2 $Terc^{-/-}$ mice compared to wild-type mice even under a control diet (Fig. 3d, e). Similarly, in the acute rapamycin treatment experiment, control 4–6 months old G2 $Terc^{-/-}$ mice showed significantly higher hepatic pS6 levels than wild-type mice in the absence of rapamycin treatment (Supplementary Fig. 6), further supporting the notion that

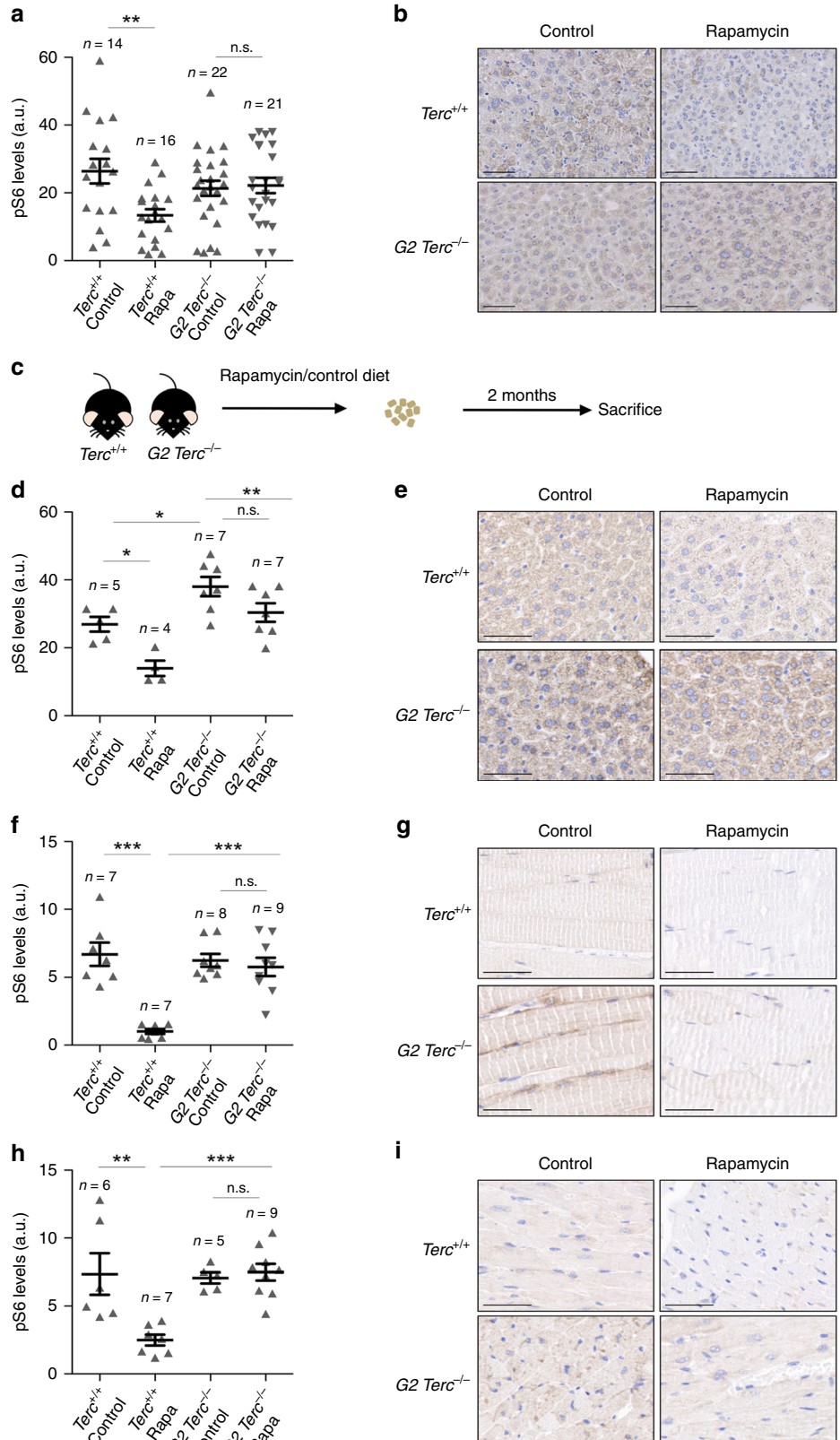

**Fig. 3 _Terc$^{-/-}$_ mice under a chronic rapamycin treatment retain S6 phosphorylation in the liver. a**, **b** Quantification (**a**) and representative images (**b**) of phosphorylated ribosomal protein S6 (pS6) expression in liver from rapamycin or control fed _Terc$^{+/+}$_ and G2 _Terc$^{-/-}$_ mice at the endpoint. **c** Two to 6 months old _Terc$^{+/+}$_ and G2 _Terc$^{-/-}$_ mice were fed rapamycin or control diet and sacrificed 2 months later. **d–i** IHC quantification (**d**, **f**, **h**) and representative images (**e**, **g**, **i**) of pS6 expression from sections of liver (**d**, **e**), skeletal muscle (**f**, **g**), and heart (**h**, **i**) of mice control and rapamycin treated for 2 months. a.u. arbitrary units. Scale bars, 100 μm. Error bars represent the SE. _n_ = number of mice. Statistical significance was determined by one-way Anova with post hoc Tukey test. *$p \leq 0.05$; **$p \leq 0.01$; ***$p \leq 0.001$; n.s. not significant. Source data are provided as a Source Data file.

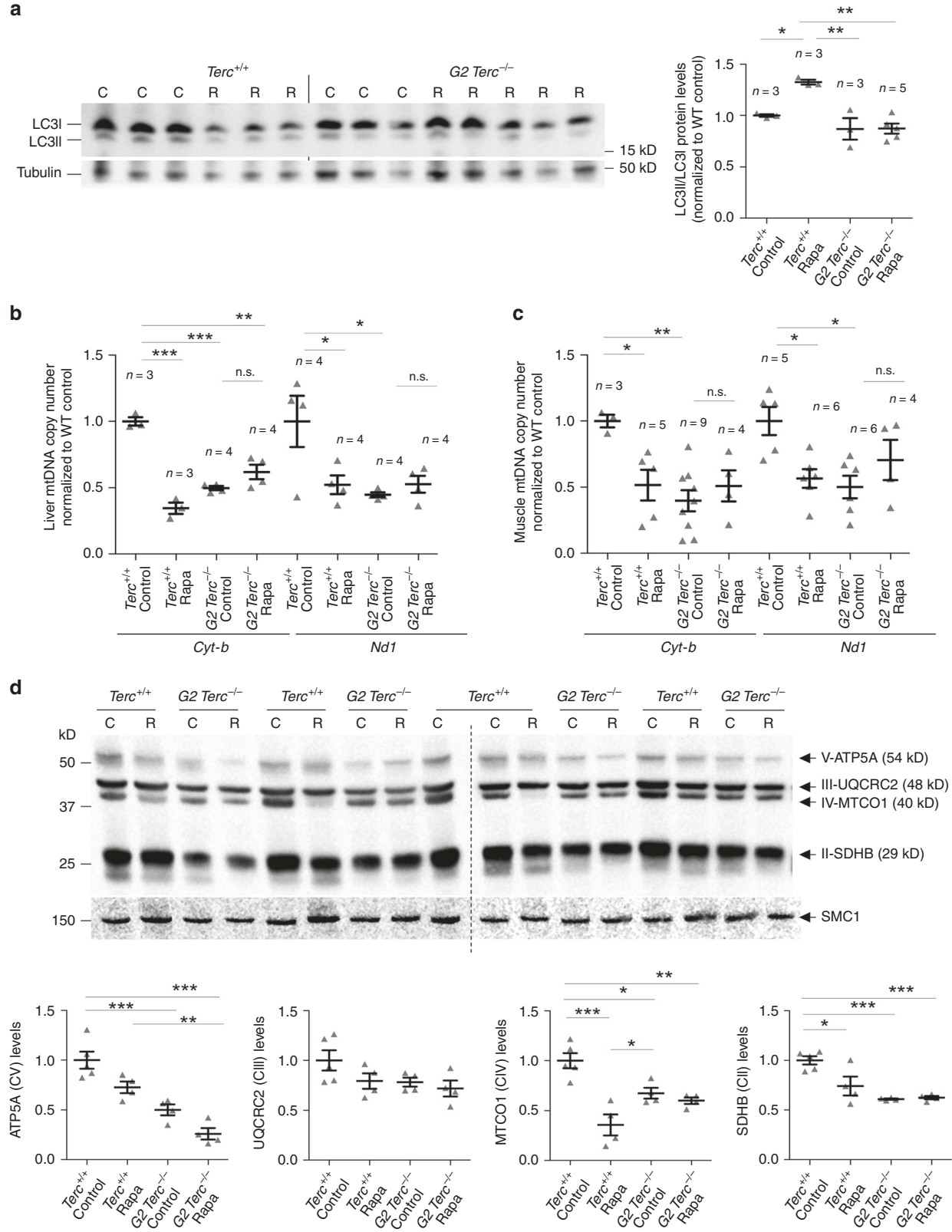

telomerase-deficient mice have an hyperactivated mTORC1 pathway. These unexpected findings open the interesting possibility that $Terc^{-/-}$ mice have a hyperactivated mTORC1 pathway in response to short telomeres and that this activation is acting as a survival pathway since inhibition of mTORC1 by chronic rapamycin treatment results in decreased longevity.

**Hyperactivation of mTOR in $Terc^{-/-}$ mice.** To address whether $Terc^{-/-}$ mice had a hyperactivated mTOR pathway, we performed RNA sequencing to analyze gene expression profiles in the liver of adult (2–6 months old) $Terc^{+/+}$ and G2 $Terc^{-/-}$ male mice fed with either rapamycin or control diet during 2 months (Fig. 3c; GEO database GSE127475). Gene-set enrichment

**Fig. 4 Worsened mitochondrial function in rapamycin-treated wild-type mice. a** Western blot representative images (left panel) and quantification (right panel) of LC3 protein levels, LC3-I and LC3-II forms (LC3-II/LC3-I ratio), and tubulin as a loading control from hepatic protein extracts from healthy male *Terc*$^{+/+}$ and G2 *Terc*$^{-/-}$ mice fed rapamycin or control diet during 2 months. **b**, **c** Mitochondrial DNA copy number as defined by analysis of the *Cyt-b* and *Nd1* mitochondrial genes in liver (**b**) and skeletal muscle (**c**) from rapamycin or control fed *Terc*$^{+/+}$ and G2 *Terc*$^{-/-}$ mice at the HEP. **d** Western blot representative images (upper panel) and quantification (lower panel) of the OXPHOS complexes V (ATP5A), III (UQCRC2), IV (MTCO1), and II (SDHB) protein levels in liver from rapamycin- or control-fed *Terc*$^{+/+}$ and G2 *Terc*$^{-/-}$ mice at the HEP. Error bars represent the SE. $n =$ number of mice. Statistical significance was determined by one-way Anova with post hoc Tukey test. *$p \leq 0.05$; **$p \leq 0.01$; ***$p \leq 0.001$; n.s. not significant. Source data are provided as a Source Data file.

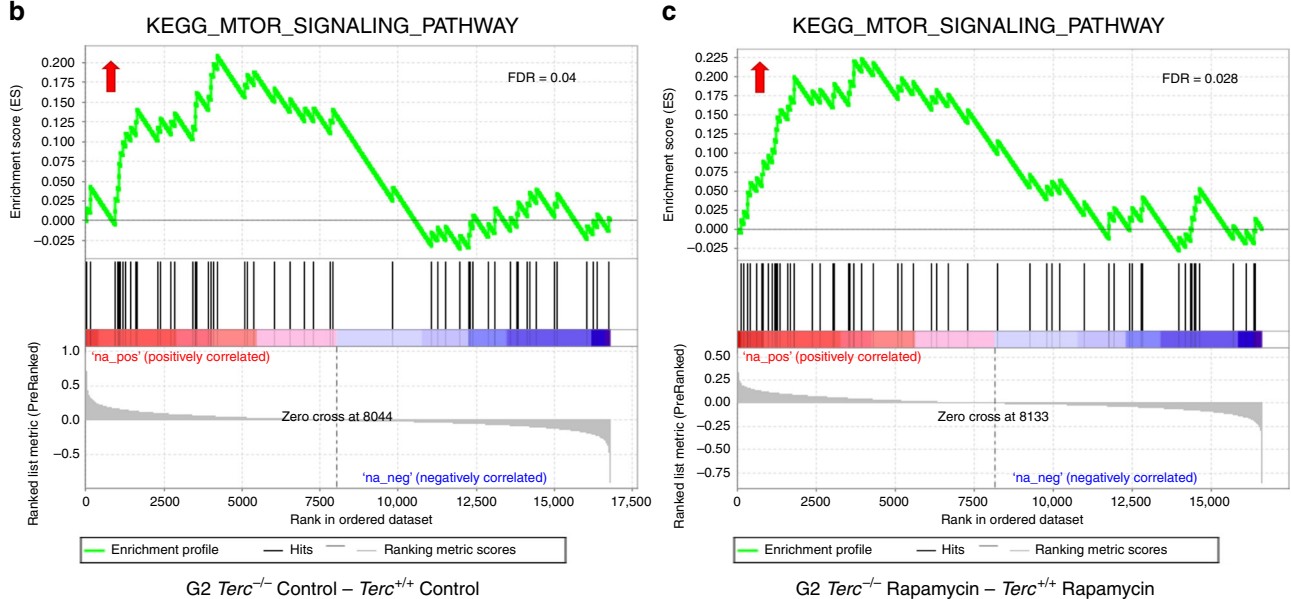

**a**

| NAME | FDR |
|---|---|
| KEGG_INSULIN_SIGNALING_PATHWAY | 0 |
| KEGG_FATTY_ACID_METABOLISM | 0 |
| KEGG_PPAR_SIGNALING_PATHWAY | 0.00180003 |
| KEGG_APOPTOSIS | 0.00249524 |
| KEGG_STEROID_BIOSYNTHESIS | 0.00238349 |
| KEGG_FOXO_SIGNALING_PATHWAY | 0.00527341 |
| KEGG_PEROXISOME | 0.00509887 |
| KEGG_PYRUVATE_METABOLISM | 0.00918704 |
| KEGG_PI3K-AKT_SIGNALING_PATHWAY | 0.02109587 |
| KEGG_MTOR_SIGNALING_PATHWAY | 0.04063314 |
| KEGG_GLYCOLYSIS_/_GLUCONEOGENESIS | 0.06458292 |
| KEGG_LYSOSOME | 0.07642359 |
| KEGG_RAS_SIGNALING_PATHWAY | 0.09683494 |
| KEGG_HIF-1_SIGNALING_PATHWAY | 0.19779144 |

**Fig. 5 Telomerase-deficient mice show an upregulated mTOR pathway.** Gene expression data obtained by RNAseq of three independent liver samples from *Terc*$^{+/+}$ and G2 *Terc*$^{-/-}$ male mice subjected to control or rapamycin diet and sacrificed after 2 months of treatment were analyzed by Gene Set Enrichment Analysis (GSEA) to determine significantly enriched gene sets. **a** Table showing significantly enriched gene sets (FDR < 0.25) between control-fed G2 *Terc*$^{-/-}$ and *Terc*$^{+/+}$. Note the upregulation of several metabolic pathways in telomerase deficient as compared to wild-type mice (for a complete list of deregulated pathways see Supplementary Table 1). The database used was the Kyoto Encyclopedia of Genes and Genomes (KEGG). **b**, **c** Gene set enrichment analysis (GSEA) plots for the mTOR pathway in G2 *Terc*$^{-/-}$ versus *Terc*$^{+/+}$ mice subjected to control (**b**) or rapamycin (**c**) diet. The red to blue horizontal bar represents the ranked list. Genes located at the central area of the bar show small differences in gene expression between the pairwise compared. At the red edge of the bar are located genes showing higher expression levels in G2 *Terc*$^{-/-}$ control fed (**b**) or G2 *Terc*$^{-/-}$ rapamycin-fed (**c**) mice; at the blue edge of the bar are located genes showing higher expression levels in *Terc*$^{+/+}$ control fed (**b**) or *Terc*$^{+/+}$ rapamycin-fed (**c**) mice. Red arrows indicated upregulation of the mTOR pathway in the pairwise comparisons. False discovery rate (FDR) is indicated in each case.

analysis (GSEA) of control fed G2 *Terc*$^{-/-}$ versus control fed *Terc*$^{+/+}$ confirmed that the mTOR pathway is upregulated in telomerase-deficient mice compared to wild-type mice (Fig. 5a; Supplementary Table 1), in agreement with higher hepatic levels

of pS6 in G2 *Terc*$^{-/-}$ mice (Fig. 3d, e; Supplementary Fig. 6). In addition, we found that other metabolic pathways related to the mTOR pathway[22] were also upregulated in G2 *Terc*$^{-/-}$ mice compared to wild-type mice, including insulin signaling,

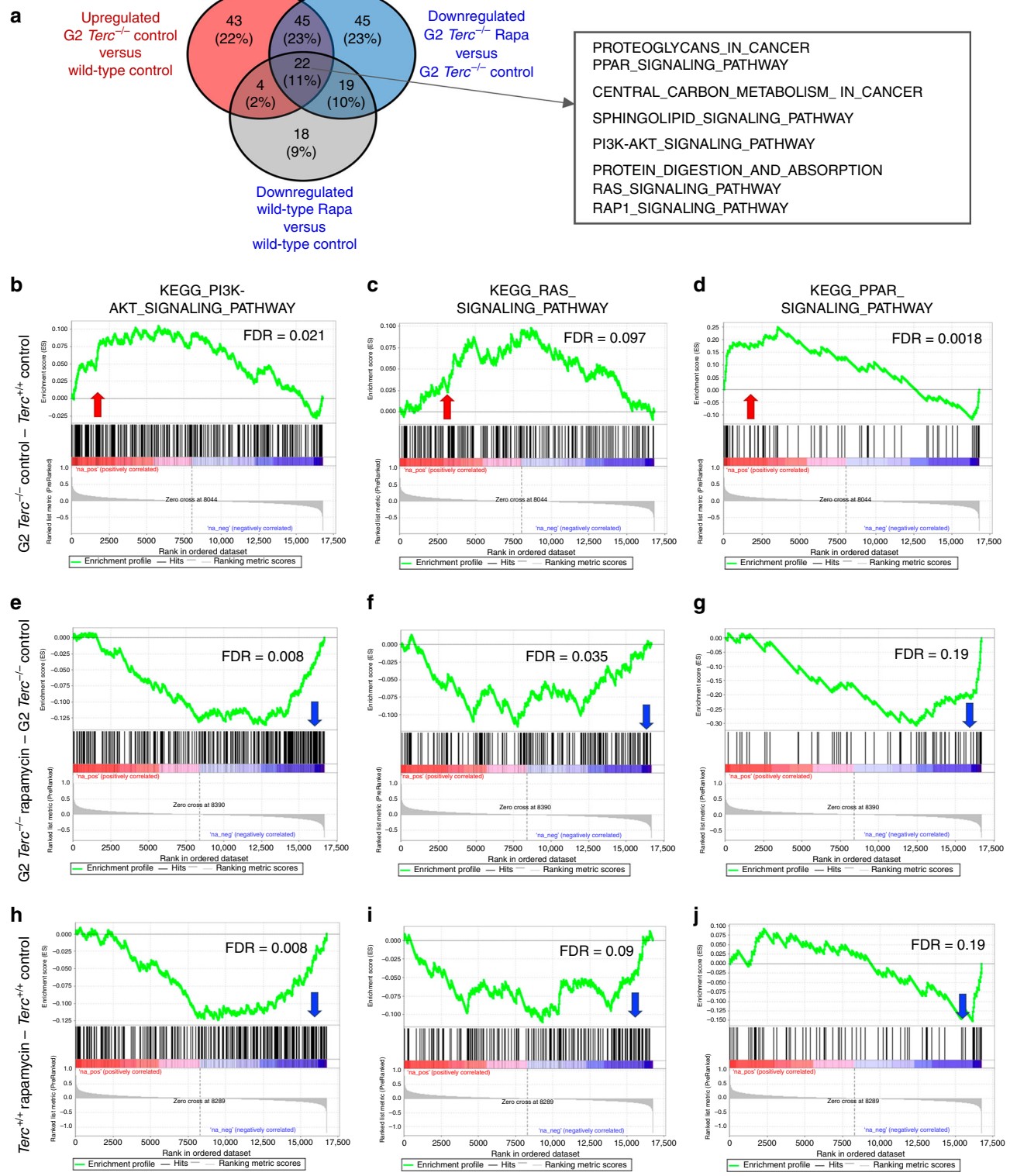

cholesterol and fatty acid biosynthesis, PPAR signaling, apoptosis, peroxisome, pyruvate metabolism, the PI3K–AKT pathway, glycolysis, lysosome, as well as the RAS and HIF-1 signaling pathways (Fig. 5a, b; Supplementary Table 1).

Further supporting hyperactivation of the mTOR pathway in $Terc^{-/-}$ mice, GSEA analysis of rapamycin-treated G2 $Terc^{-/-}$ versus rapamycin-treated $Terc^{+/+}$ also showed upregulation of the mTOR pathway in the telomerase-deficient mice compared to wild-type mice, even under the rapamycin treatment (Fig. 5c; Supplementary Table 2).

Next, we compared all pathways (KEGG database) found upregulated in control fed G2 $Terc^{-/-}$ versus control fed $Terc^{+/+}$ to all the pathways found downregulated both in rapamycin-treated G2 $Terc^{-/-}$ versus control-fed G2 $Terc^{-/-}$ and in rapamycin-treated wild type versus control-fed wild-type (Fig. 6a). We detected a total of 22 common pathways at the intersection of

**Fig. 6 Hyperactivation of mTOR-related metabolic pathways in $Terc^{-/-}$ mice.** Gene expression data obtained by RNAseq of three independent liver samples from $Terc^{+/+}$ and G2 $Terc^{-/-}$ male mice subjected to control or rapamycin diet and sacrificed after 2 months of treatment were analyzed by Gene Set Enrichment Analysis (GSEA) to determine significantly enriched gene sets. **a** Venn diagram showing the overlapping pathways found significantly upregulated in control fed G2 $Terc^{-/-}$ versus $Terc^{+/+}$ and those found significantly downregulated in rapamycin-treated G2 $Terc^{-/-}$ versus control fed G2 $Terc^{-/-}$ and in rapamycin-treated wild type versus control-fed wild type. Twenty-two common pathways were detected at the intersection of the three comparisons. Several pathways related to the mTOR pathway are listed in the box. **b–j** GSEA plots for the indicated pathways in liver samples. **b–d** G2 $Terc^{-/-}$ versus $Terc^{+/+}$ control-fed mice. (**e-g**) G2 $Terc^{-/-}$ rapamycin-fed versus G2 $Terc^{-/-}$ control-fed mice. **h–j** $Terc^{+/+}$ rapamycin-fed versus $Terc^{+/+}$ control-fed mice. The red to blue horizontal bar represents the ranked list. Genes located at the central area of the bar show small differences in gene expression between the pairwise compared. At the red edge of the bar are located genes showing higher expression levels and at the blue edge of the bar are located genes showing lower expression levels. Red and blue arrows indicated upregulation and downregulation, respectively, of the pathway in the pairwise comparisons. Samples correspond to livers of three independent G2 $Terc^{-/-}$ or $Terc^{+/+}$ male mice sacrificed after 2 months of control or rapamycin feeding. False discovery rates (FDR) are indicated.

these comparisons, including several mTOR-related pathways (Fig. 6a; Supplementary Tables 1–4). Interestingly, GSEA analysis showed that control fed G2 $Terc^{-/-}$ also had a significant upregulation of the mTOR upstream regulator pathways PI3K–AKT and Ras signaling compared to wild-type controls (Fig. 6a–c; Supplementary Table 1)[22,49–52], further supporting hyperactivation of the mTOR pathway in $Terc^{-/-}$ mice. Furthermore, GSEA analysis showed that control fed G2 $Terc^{-/-}$ also had a significant upregulation of the mTOR downstream PPAR signaling pathway compared to the wild-type controls (Fig. 6a, d; Supplementary Table 1)[53–56].

Interestingly, rapamycin treatment leads to downregulation of PI3K–AKT, Ras and PPAR signaling pathways in both G2 $Terc^{-/-}$ and $Terc^{+/+}$ compared to control-fed counterparts (Fig. 6a, e–j; Supplementary Tables 3 and 4).

Together, these results demonstrate that $Terc^{-/-}$ mice with short telomeres have higher basal activation levels of the mTOR pathway, as well as other related metabolic pathways, suggesting an increased metabolism. In turn, this increased metabolism might be a survival response to telomerase deficiency and presence of short telomeres, as supported by the fact that inhibition of these metabolic routes by rapamycin negatively impacts in the survival of telomerase-deficient mice.

**Reduced survival in $Terc^{-/-}$ $S6K1^{-/-}$ mice.** Our results suggest a previously unnoticed role of the mTOR pathway in promoting organismal survival in the absence of telomerase activity and in the presence of short telomeres. To further demonstrate this notion using mouse genetics, we deleted the downstream target of mTOR, the ribosomal protein S6 kinase 1 (S6K1), known to positively regulate protein synthesis and cell growth[57,58]. We generated a double-knockout mouse model for $S6k1$ and $Terc$ by crossing $S6k1^{-/-}$ mice[59] with $Terc^{+/-}$ heterozygous mice[14] to obtain G1 $Terc^{-/-}$ $S6k1^{-/-}$ and G1 $Terc^{-/-}$ $S6k1^{+/+}$. Successive $Terc^{-/-}$ $S6k1^{+/-}$ intercrosses generated the G2–G3 $Terc^{-/-}$ $S6k1^{-/-}$ and G2–G3 $Terc^{-/-}$ $S6k1^{+/+}$ experimental groups.

Deletion of $S6k1$ was previously known to lead to a 19% increase in median longevity in female mice[32]. In agreement with this, we observed an 8% increase in median survival of $Terc^{+/+}$ $S6k1^{-/-}$ females compared to wild-type counterparts (Fig. 7a), while no differences in longevity were found between the $Terc^{+/+}$ $S6k1^{-/-}$ and $Terc^{+/+}$ $S6k1^{+/+}$ males (Fig. 7b) or when considering both sexes together (Fig. 7c). With increasing generations of $Terc^{-/-}$ we observed a negative effect of S6K1 abrogation, and G3 $Terc^{-/-}$ $S6k1^{-/-}$ female mice showed a significant 20% decrease in median lifespan compared to G3 $Terc^{-/-}$ $S6k1^{+/+}$ counterparts (Fig. 7a). A similar trend was observed in male mice although the difference did not reach statistical significance (Fig. 7b). These findings demonstrate that S6K1 is a survival signal in the context of telomerase deficiency as

S6k1 deficiency further decreases longevity in G3 $Terc^{-/-}$ females. The fact that the decreased survival was observed with increasing mouse generations of $Terc^{-/-}$ mice indicates that it is the presence of short telomeres what causes deleterious effects of S6K1 inhibition in these mice.

We analyzed the levels of pS6 in the livers of these mouse cohorts. In agreement with previous work, S6K1 deficiency does not affect the pS6 levels in either $Terc^{+/+}$ or in G2 $Terc^{-/-}$ backgrounds (Supplementary Fig. 7A)[59,60]. To address whether the effect of S6K1 deficiency on mouse survival was dependent on telomere maintenance, we measured telomere length in these mouse cohorts by performing Q-FISH analysis in mice at the human endpoint. To avoid age-related variations in telomere length within each experimental group, we compared $S6k1^{+/+}$ and $S6k1^{-/-}$ within each $Terc^{-/-}$ generation groups that died at same age, namely 22–26 months old in the case of $Terc^{+/+}$ mice, 13–18 month old in G2 $Terc^{-/-}$ mice, and 12–14 month old in the case of G3 $Terc^{-/-}$ mice. Telomere length was determined in tissue sections from the intestine and the liver (Supplementary Fig. 7B–C). The results show that S6K1 deficiency does not affect mean telomere length neither in a highly proliferative tissue as the intestine nor in a post-mitotic tissue such as the liver, indicating that S6K1 function in mouse survival is independent of telomere maintenance. As expected, successive generations of $Terc^{-/-}$ mice present progressively shorter telomeres (Supplementary Fig. 7B, C). These findings demonstrate that the increased longevity of $S6k1^{-/-}Terc^{+/+}$ female mice is not due to a better preservation of telomere length during aging mediated by telomerase. Similarly, we did not see that S6K1 deficiency aggravated telomere shortening in the context of $Terc^{-/-}$ mice (Supplementary Fig. 7B, C). Histopathological analysis at human endpoint of the different mouse cohorts showed a decreased lymphoma incidence in $S6k1^{-/-}Terc^{+/+}$ mice compared to $S6k1^{+/+}Terc^{+/+}$ mice while no difference was observed in the frequency of sarcoma (Supplementary Fig. 7D–E). These results are in agreement with the reduced lymphoma observed in $Terc^{+/+}$ mice treated with rapamycin (Fig. 1f), confirming a protective role of mTORC1 pathway inhibition in lymphoma development. We did not observe significant difference in the incidence of intestinal degenerative pathologies developed by late generation (G2–G3) $Terc^{-/-}$ mice independently of S6K1 status (Supplementary Fig. 7F), also in agreement with the results obtained in rapamycin-treated G2 $Terc^{-/-}$ (Fig. 1h).

In summary, these findings provide genetic support for our observation that chronic rapamycin treatment has deleterious effects in the context of telomerase deficiency and presence of short telomeres.

## Discussion

Telomerase deficiency and presence of telomeres shorter than normal have been shown to induce premature loss of the

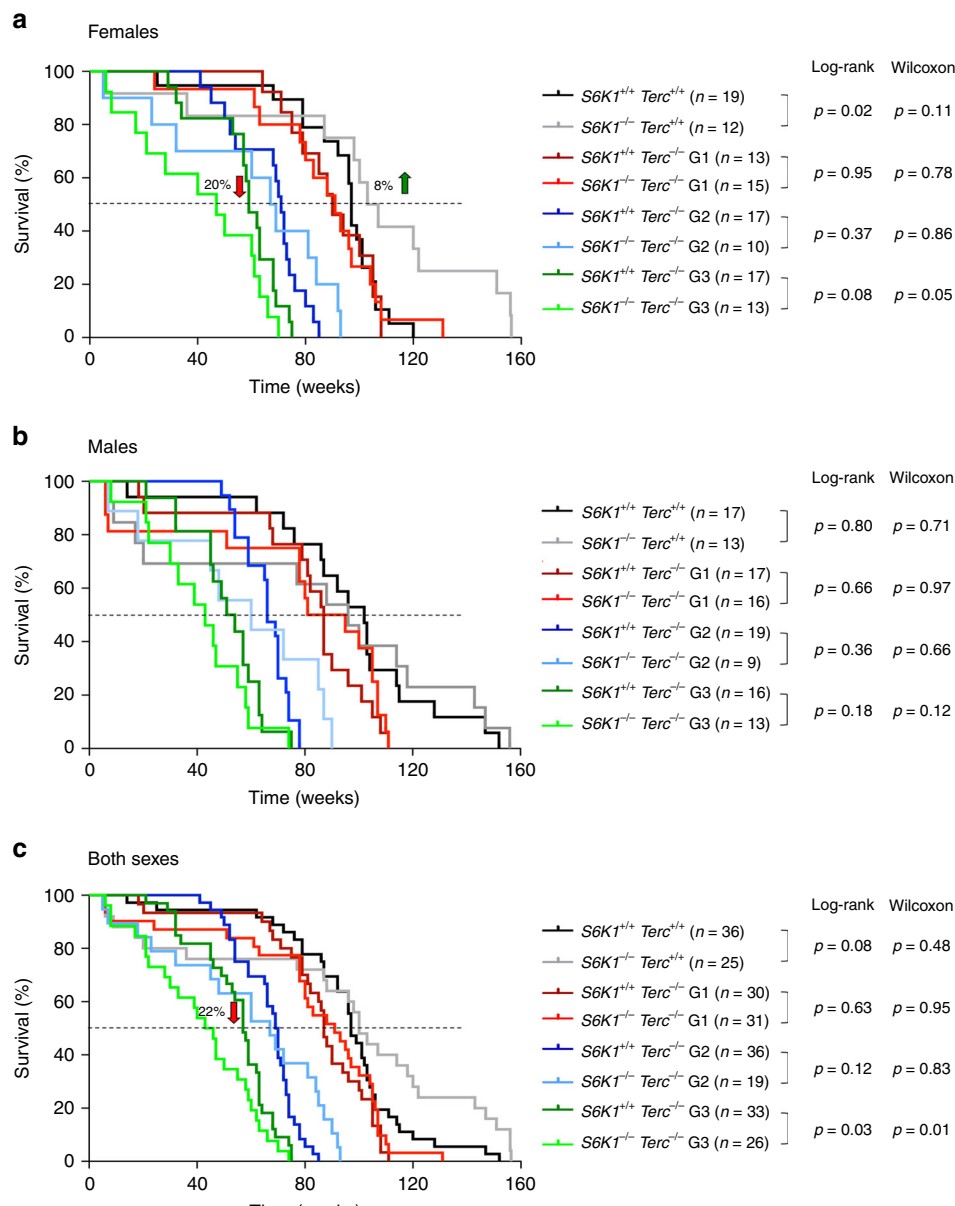

**Fig. 7 Reduced survival in late generation _Terc_<sup>−/−</sup> S6K1<sup>−/−</sup> mice.** Kaplan–Meier survival curves of _S6k1_$^{+/+}$ _Terc_$^{+/+}$, _S6k1_$^{-/-}$ _Terc_$^{+/+}$, G1–G3 _S6k1_$^{+/+}$ _Terc_$^{-/-}$, and G1–G3 _S6k1_$^{-/-}$ _Terc_$^{-/-}$ mice. **a** Females; the percentage of increase in median survival of _S6k1_$^{-/-}$ _Terc_$^{+/+}$ mice relative to that of _S6k1_$^{+/+}$ _Terc_$^{+/+}$ is indicated (green arrow); the percentage of decrease in median survival of G3 _S6k1_$^{-/-}$ _Terc_$^{-/-}$ mice relative to that of G3 _S6k1_$^{+/+}$ _Terc_$^{-/-}$ is indicated (red arrow). **b** Males. **c** Both sexes; the percentage of decrease in median survival of G3 _S6k1_$^{-/-}$ _Terc_$^{-/-}$ mice relative to that of G3 _S6k1_$^{+/+}$ _Terc_$^{-/-}$ is indicated (red arrow). _n_ = number of mice. Statistical significance was determined by the log-rank test and by the Gehan–Breslow–Wilcoxon test. The _p_ values are indicated. Source data are provided as a Source Data file.

regenerative capacity of tissues and premature aging pathologies both in mice and humans[17,18]. In humans, mutations in telomerase or telomere-maintenance genes are at the origin of the so-called telomere syndromes, which are characterized by the presence of critically short telomeres. These diseases include Hoyeral–Hreidarsson syndrome, dyskeratosis congenita, pulmonary fibrosis, aplastic anemia, and liver fibrosis[17,18]. Although telomerase-activation strategies are being studied as potential treatment for these diseases[61–63], currently there are still no curative therapeutic options for these patients.

The mechanistic target of rapamycin or mTOR is a serine/threonine protein kinase of the PI3K family that functions as a master regulator of cell growth and metabolism in response to nutrients and hormonal cues. Inhibition of mTOR pathway has

been shown to extend lifespan and delay age-related pathologies across many different species, from yeast to mice[64]. Rapamycin is the first mTOR inhibitor to be isolated and has extensively been studied in its ability to extend lifespan, decrease cancer, and have immunosuppressant effects[29,30,35].

Here, we set to address a functional interaction between the mTOR pathway and telomeres, by testing whether mTOR inhibition could also extend the lifespan of telomerase-deficient mice with short telomeres[29–31,65] as this could serve as a proof of principle that inhibition of the mTOR pathway may be a therapeutic approach for the so-called telomere syndromes. Unexpectedly, we found that rapamycin treatment decreases the longevity of telomerase-deficient mice with short telomeres, in marked contrast to very significant lifespan extension in the case

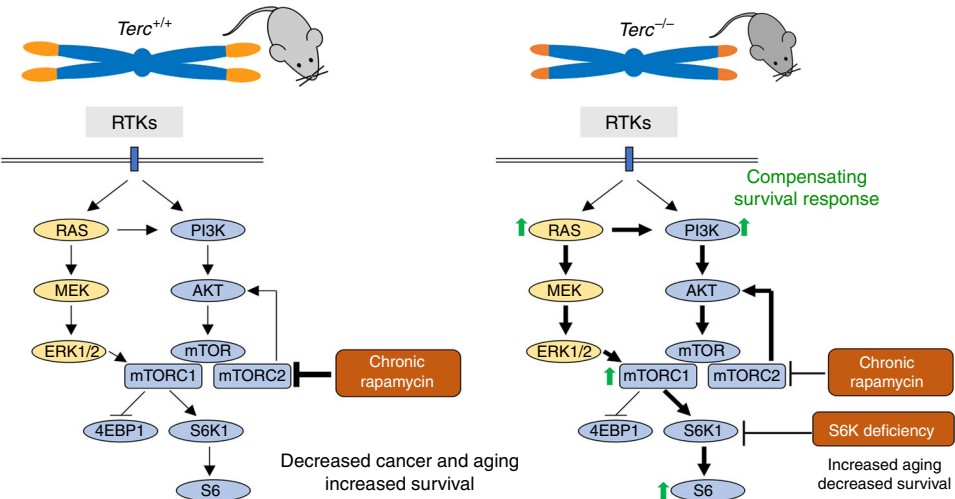

**Fig. 8 Chronic rapamycin treatment abolishes the endogenous survival response of *Terc*⁻/⁻ mice.** The mechanistic target of the rapamycin (mTOR) pathway is a central regulator of cell growth and metabolism. A variety of upstream signals regulate mTOR activity, including growth factors and nutrients. Activation of receptor tyrosine kinases (RTKs) by growth factors activates the PI3K/AKT pathway as well as the RAS and MEK/ERK pathway, which promote mTORC1 activation. Chronic rapamycin treatment inhibits mTORC1 activity and in some cell types also mTORC2 activity. In telomerase-deficient mice with short telomeres, the G2 *Terc*⁻/⁻ mice, the RAS and PI3K/AKT pathways are upregulated concomitantly with increased mTORC1 activity and increased levels of the mTORC1 downstream target pS6. The results shown here support a model in which hyperactivation of these metabolic-related pathways as the consequence of telomerase deficiency and short telomeres constitute a compensatory survival mechanism in response to the telomeric defects. In agreement with this, inhibition of the mTOR pathway by chronic rapamycin treatment or by genetic deletion of S6K1 is deleterious for telomerase-deficient mice.

of wild-type controls. Of interest, wild-type mice treated with rapamycin showed an increase that is higher than that reported by other authors using the same diet in genetically heterogeneous mice[31]. These discrepancies may be due to different genetic backgrounds or to different housing conditions[66] and/or to the fact that we initiated the rapamycin treatment at 3 months of age instead of 9 months of age[31].

We rule out here that decreased survival of rapamycin-fed *Terc*⁻/⁻ mice is due to effects of rapamycin aggravating telomere shortening or telomere aberrations in the absence of telomerase. Intriguingly, we found that rapamycin increased telomere fragility in wild-type mice, which could be suggestive of increased replicative stress associated to rapamycin treatment, an observation which warrants future research.

The fact that rapamycin treatment was deleterious in the context of short telomeres and telomerase deficiency suggests that the mTOR pathway maybe acting as a pro-survival pathway in this setting. In support of this, we find that *Terc*⁻/⁻ mice show a hyperactivated mTORC1 pathway. RNA sequencing further demonstrated hyperactivation of the mTOR pathway and of related metabolic pathways in *Terc*⁻/⁻ mice. Among the upregulated pathways, we find insulin signaling, cholesterol and fatty acid biosynthesis, PPAR signaling, peroxisome, pyruvate metabolism, the PI3K–AKT pathway, glycolysis, lysosome, RAS signaling, and HIF-1 signaling pathway. Of interest, we also previously observed upregulation of the PI3K/AKT pathway in the skin of *Terc*⁻/⁻ mice with extremely short telomeres[67]. Upregulation of these pathways in mice with short telomeres is consistent with the need of higher energy consumption in these mice, lower mitochondria content[48], and the notion that they are acting as survival pathways in the context of telomerase deficiency and presence of short telomeres (see model in Fig. 8). Indeed, rapamycin treatment significantly inhibited the PI3K/AKT and Ras pathways upstream of mTOR in both wild type and telomerase-deficient mice, indicating that the inhibition of these survival pathways is underlying the detrimental effects of rapamycin treatment in the setting of short telomeres (see model in Fig. 8).

We further validate this notion using mouse genetics, and demonstrate here that female mice doubly deficient in telomerase and S6 kinase 1 of third generation (G3) also show a decreased survival in contrast to increased survival of S6K1-deficient females compared to wild-type controls (see model in Fig. 8). The fact that the decreased survival was only observed in the third generation of the telomerase-deficient mice indicates that it is not the lack of active telomerase but the presence of very short telomeres what causes the synthetic lethality with S6K1 deficiency. While chronic rapamycin affects negatively already in G2 *Terc*⁻/⁻, the effects of S6K1 deficiency become apparent in G3. This discrepancy might be explained by different telomere length due to different genetic backgrounds of both mouse colonies and by the fact that S6K1 and S6K2 present functional compensation and redundancy[59,60].

Finally, the fact that mTORC1 activity has been found elevated in the livers of old mice[68] suggests that this may be a general phenomenon associated not only with telomere-induced aging but also with physiological aging.

## Methods
**Mice, husbandry, and diet preparation.** All mice were generated and maintained at the Animal Facility of the Spanish National Cancer Research Centre (CNIO) under specific pathogen-free conditions in accordance with the recommendation of the Federation of European Laboratory Animal Science Associations (FELASA). C57BL/6 *Terc*⁺/⁻ heterozygous female and male mice[14] were intercrossed to obtain *Terc*⁺/⁺ and first-generation (G1) *Terc*⁻/⁻ litters. G1 *Terc*⁻/⁻ mice were then intercrossed to obtain second-generation (G2) *Terc*⁻/⁻ offspring. All the mice included in survival studies were fed control chow from the weaning onwards. At 3 months of age, mice were divided in two groups, one of them continued with control chow and the other were changed to chow-containing encapsulated rapamycin at 42 ppm (mg of drug per kg of food). All mice were followed until their endpoint. Rapamycin was microencapsulated by Rapamycin Holdings Inc. (San Antonio, Texas) using a spinning disk atomization coating process with the enteric coating material Eudragit S100 (Rohm Pharma)[31]. This coating material increases the fraction of rapamycin that survives the food preparation process; moreover, being water soluble only, it allows the drug to be released in the small intestine rather than in the stomach, then increasing its absorption. Encapsulated rapamycin was then incorporated into 5LG6 mouse chow (TestDiet, London, UK).

Food and water were provided ad libitum and measurements of the body weight were performed monthly.

For studying the molecular hepatic response to chronic rapamycin treatment in healthy young mice, male and female *Terc*[+/+] and G2 *Terc*[−/−] mice were given rapamycin diet or control diet and sacrificed after the time indicated. To ensure that all mice were in the same nutritional conditions at the time of sacrifice, we fasted them overnight and refed them (ad libitum, rapamycin or control chow) for 3 h before sacrifice.

The response to an acute rapamycin treatment was performed by intraperitoneal injection (i.p.) of rapamycin (2 mg/kg body weight) in 4–6 months old *Terc*[+/+] and G2 *Terc*[−/−] males followed by mouse sacrifice 2 h later. To measure the basal activation of the mTOR pathway in absence of treatment, these mice were fasted overnight and refed for 1 h (ad libitum) before sacrifice, to ensure similar nutritional conditions.

To generate double-knockout mice for *S6k1* and *Terc*, *S6k1*[−/−] mice[59] were first crossed with *Terc*[+/−] heterozygous mice[14] to obtain G1 *Terc*[−/−] *S6k1*[+/+], G1 *Terc*[−/−] *S6k1*[+/−], and G1 *Terc*[−/−] *S6k1*[−/−] litters. G1 *Terc*[−/−] *S6k1*[+/−] mice were then intercrossed to generate G2 *Terc*[−/−] *S6k1*[+/+], G2 *Terc*[−/−] *S6k1*[+/−], and G2 *Terc*[−/−] *S6k1*[−/−] litters. G3 *Terc*[−/−] *S6k1*[+/+] and G3 *Terc*[−/−] *S6k1*[−/−] mice were then obtained by mating G2 *Terc*[−/−] *S6k1*[+/−] female and male mice. Food (Harlan Laboratories) and water were provided ad libitum.

All animal experiments were approved by the Ethical Committee (CEIyBA) (IACUC.040-2014, CBA_20_2014) and performed in accordance with the guidelines stated in the International Guiding Principles for Biomedical Research Involving Animals, developed by the Council for International Organizations of Medical Sciences (CIOMS).

**Immunohistochemistry analysis**. Tissue samples were fixed in 10% buffered formalin, dehydrated, embedded in paraffin wax, and sectioned at 2.5 mm. Tissue sections were deparaffinized in xylene and re-hydrated through a series of graded ethanol until water and then stained with hematoxylin and eosin for pathological examination. The classification of the intestinal pathologies was performed by a trained pathologist in a blinded manner. Mild intestinal lesions are characterized by multifocal epithelia and glandular atrophy that affect up to 20% of the tissue. Medium and severe intestinal lesions show multifocal areas in the mucosa lacking glands and the presence of degenerative epithelial and glandular cystic hyperplasia that affect between 21% and 60% or > 61% of the tissue, respectively.

Immunohistochemistry was performed on deparaffinized tissue sections processed with 10 mM sodium citrate (pH 6.5) cooked under pressure for 2 min. Slides were washed in water, then in Buffer TBS-0.5% Tween 20, blocked with peroxidase, washed with TBS-0.5% Tween 20, and blocked with fetal bovine serum. Intestine or skeletal muscle sections were incubated with the following primary antibodies: mouse monoclonal to phospho-histone H2AX (ser139) (1:15,000; JBW301, Millipore, Cat#05-636), rat monoclonal to p21 (1:10; 291H/B5), rat monoclonal to p53 (1:100; POE316 A/E9), rabbit polyclonal to phospho-histone H3 (1:500; Abcam, Cat#ab5168), rat monoclonal to p19 ARF (1:50; 5-C3-1, Santa Cruz Biotechnology, Cat#sc-32748), or rabbit polyclonal to AC3 cleaved-caspase 3 (Asp175) (1:300; Cell Signaling Technology, Cat#9661). Liver, heart, and skeletal muscle sections were incubated with a rabbit polyclonal anti-phospho-S6 ribosomal protein (Ser240/244) (1:500; Cell Signaling Technology, Cat#2215). Slides were then incubated with secondary antibodies conjugated with peroxidase from DAKO. Sections were lightly counterstained with hematoxylin and analyzed by light microscopy. The number of positive cells for phospho-H2AX, p21, p53, and AC3 was quantified by eye using an Olympus AX70 microscope. The percentage of tissue area positive for phospho-S6 was quantified with the AxioVision microscope software from Carl Zeiss.

**Immunofluorescence analyzes on tissue sections**. For immunofluorescence analyzes, intestine sections were fixed in 10% buffered formalin (Sigma) and embedded in paraffin. After deparaffinization and citrate antigen retrieval, sections were permeabilized with 0.5% Triton in PBS and blocked with 1% BSA and 10% Australian FBS (GENYCELL) in PBS. The antibodies were applied overnight in antibody diluents with background reducing agents (Invitrogen). Intestinal foci of replication protein A (RPA) were detected with a rat polyclonal antibody anti-RPA 32 (1:200; 4E4 Cell Signaling Technology) and further incubated with 555 Alexa Fluor goat anti-rat antibody. A double immunofluorescence using primary antibodies against 53BP1 (1:500; Novus Biologicals) and TRF1 (1:500; homemade rat monoclonal) was performed to assay for telomeric DNA damage. Slides were then incubated with 488 Alexa Fluor goat anti-rabbit and 555 Alexa Fluor goat anti-rat secondary antibodies. Immunofluorescence images were obtained using a confocal ultraspectral microscope (Leica TCS-SP5). Quantifications were performed with Definiens software.

**Rapamycin quantification in liver and blood samples**. Liver samples were mechanically homogenized in water with 2.8 mm ceramic bead in a Precellys® Tissue Homogenizer. Homogenate was incubated in ultrasonic bath 10 min and centrifuged at 15,000*g* for 5 min. Blood samples were centrifuged twice at 500*g* for 10 min. The supernatants were placed into a 96-well solid phase extraction plate and proteins were precipitated with 4 volumes of 0.1% formic acid/acetonitrile.

Vacuum was applied and sample recovered into a 96-well plate and analyzed by specific LC-MS/MS method in MRM mode. MS analysis was conducted using a Qtrap 5500 mass spectrometer (AB Sciex). The turboinspray source (TIS) was operated in positive ionization mode. Instrument control and data analysis were performed using Analyst® 1.6.2 application software from Applied Biosystems. Mass spectrometry parameters optimization of rapamycin was carried out by a standard stock dilution (100.0 ng/mL) in methanol through direct infusion into the mass spectrometer. Data from rapamycin were first acquired in full scan from the range between $m/z$ 50 and 1000 in order to identify the most suitable parent ion for MS/MS experiments. The sodium adduct [M+Na]+ at $m/z$ 936.348 was selected as the parent ion for rapamycin and fragmented. The MS/MS parent ion was still preserved as a parent in the MS2 spectrum and it was together with two other daughter ions at $m/z$ 409.300 and 453.300. Chromatographic separation was carried out on an Agilent 1100 series LC system consisting of a G1312A binary pump, a G1379A degasser, and an ALSG1330B refrigerated autosampler. Isocratic mobile phase composed of 80% acetonitrile in water mixture containing 0.1% of formic acid at a flow rate of 0.6 mL/min was used. As sample injection volume of 20.0 μL was used, and total analytical run time was 6.0 min. LC-MS/MS chromatographic separation was achieved on a Phenomenex Gemini 3 μm C18 110 Å analytical column. The column oven temperature was maintained at 40 °C. Twenty microliters were injected on LC-MS/MS. Ionspray source temperature 350 °C and ionspray voltages 5500 V were optimized. Mass spectrometry data were acquired in positive ion mode and processed using Analyst software (version 1.6.2, AB Sciex). An LC-MS/MS analysis, rapamycin was eluted at retention time 3.2 min. The curtain gas (CUR) was at 40.0 psi, the nebulizer source gas 1 at 40.0 psi, and the turbo ion source gas 2 at 45.0 psi was utilized. Declustering potential 105.0 V and entrance potential 10.0 V were optimized. The collision gas pressure was medium. The collision energy 75.0 V for rapamycin and the collision cell exit potential 10.0 V. Known concentration standards were prepared by spiking control liver and plasma homogenates with the rapamycin at different concentrations.

**Metabolic measurements**. To perform GTT and ITT, mice were i.p. injected with 2 g of glucose/kg of body weight and 0.75 U insulin/kg of body weight (Eli Lilly; Humalog Insulin), respectively. In the case of GTT mice were previously fasted for 16 h. Blood glucose levels were measured with the StatStrip® glucose meters from Nova biomedical at the indicated times after injection. Plasma IGF1, insulin, and IL-6 levels were determined by ELISA with m/rIGF-1 ELISA mediagnost; Ultra Sensitive Mouse Insulin ELISA kit (Crystal Chem) and Quantikine ELISA, mouse IL-6 (R&D systems), respectively. ATP levels in liver lysates were measured by colorimetric assay (OD 570 nm) with the ATP Assay Kit (Abcam, ab83355).

**Mitochondrial copy number**. Relative mtDNA content were obtained by the comparative Ct method[48]. Briefly, we measured by qPCR the mtDNA genes *Cytb* and *Nd1* and the nuclear DNA (nucDNA) gene *H19*. The mtDNA averaged Ct values were subtracted from the nucDNA averaged Ct values obtaining the ΔCt (ΔCt = nucDNA Ct−mtDNA Ct). The relative mitochondrial DNA content was calculated by raising 2 to the power of ΔCt and then multiplying by 2 (mtDNA copy number = 2 × 2^ΔCt). Primers were used as follows: CYTB-F 5′-ATTCCTT-CATGTCGGACGAG-3′, CYTB-R 5′-ACTGAGAAGCCCCCTCAAAT-3′, ND1-F 5′-AATCGCCATAGCCTTCCTAACAT-3′, ND1-R 5′-GGCGTCTGCAAATGGT TGTAA-3′, H19-F 5′-GTACCCACCTGTCGTCC-3′, H19-R 5′-GTCCACGAGAC CAATGACTG-3′.

**Quantitative fluorescence in situ hybridization (Q-FISH) in tissue samples**. For Q-FISH, paraffin-embedded intestine or liver sections were deparaffinized and fixed with 4% formaldehyde, followed by digestion with pepsine/HCl and a second fixation with 4% formaldehyde. Slides were dehydrated with increasing concentrations of EtOH (70%, 90%, 100%) and incubated with the telomeric (TTAGGG) probe labeled with Cy3 at 85 °C for 3 min followed by 2 h at room temperature in a wet chamber. The slides were extensively washed with 50% formamide and 0.08% TBS-Tween 20 (ref. [9]). Confocal microscopy was performed at room temperature with a laser-scanning microscope (TSC SP5) using a Plan Apo 63Å-1.40 NA oil immersion objective (HCX). Maximal projection of z-stack images generated using advanced fluorescence software (LAS) were analyzed with Definiens XD software package. The DAPI images were used to detect telomeric signals inside each nucleus.

To analyze telomere length in a longitudinal manner in individual mice, white blood cells were obtained from mice of both genotypes at different time points during treatment (4.5, 6–7, 12–15, and 15–20 months of age, as well as at their endpoint). High throughput (HT)-Q-FISH on peripheral blood leukocytes was done using 120–150 μL of blood as described above[38]. Confocal images were captured using the Opera High-Content Screening system (Perkin Elmer). Telomere length values were analyzed using individual telomere spots (>10,000 telomere spots per sample).

**Fluorescence in situ hybridization in primary MEFs**. Primary embryonic fibroblasts (MEFs) were isolated from 13.5-day-old wild type or G2 *Terc*[−/−] embryos according to standard protocols. Briefly, after removal of the head and organs the whole embryo was minced and rinsed in ice-cold PBS, incubated in trypsin/EDTA

(Gibco, Grand Island, NY) before dissociating in complete medium. Cells were plated in 10 cm plates containing DMEM plus 10% FBS and incubated at 37 °C. Colcemide (Gibco) was added at a concentration of 0.1 μg/mL during the last 4 h and cells were harvested by centrifugation. After hypotonic swelling in sodium citrate (0.03 M) for 25 min at 37 °C, the cells were fixed in methanol:acetic acid (3:1). After two additional changes of fixative, the cell suspension was dropped on wet, clean slides and dried overnight. FISH was performed as described above[69]. Metaphases images were captured with the Leica LAS-X software (3.4.2. version). The incidence of chromosomal aberrations per metaphase was determined by eye.

**Immunoblotting.** Protein extracts were obtained using Nuclear Cytosolic Fractionation Kit (Biovision) or RIPA extraction buffer and protein concentration was determined using the Bio-Rad DC Protein Assay (Bio-Rad). Up to 20 μg of protein per extract were separated in SDS–polyacrylamide gels by electrophoresis. After protein transfer onto nitrocellulose membrane (Whatman), the membranes were incubated with the following primary antibodies: anti-S6 ribosomal protein (1:1000; Cell Signaling Technology, Cat#2217), anti-phospho-S6 ribosomal protein (Ser 240/244) (1:1000; Cell Signaling Technology, Cat#2215), anti LC3 (1:500; Cell Signaling Technology, Cat#2775), anti-β-actin (Sigma), anti-SMC1 (1:2000; Bethyl), anti-total OXPHOS (1:500; Total OXPHOS Rodent WB Antibody Cocktail, Cell Signaling Technology, Cat#110413). Antibody binding was detected after incubation with a secondary antibody coupled to horseradish peroxidase using chemiluminescence with ECL detection KIT (GE Healthcare). Protein-band intensities were measured with ImageJ software.

**Gene expression analysis.** Total RNA was extracted from liver of $Terc^{+/+}$ and G2 $Terc^{-/-}$ male mice fed rapamycin or control diet during 2 months. RNA samples from three independent mice per condition were analyzed. Total RNA (1 μg) was used for the RNAseq experiment. The PolyA+ fraction was purified and randomly fragmented, converted to double stranded cDNA, and processed through subsequent enzymatic treatments of end-repair, dA-tailing, and ligation to adapters with the NEBNext Ultra II Directional RNA Library Prep Kit for Illumina (NEB, Cat. No. E7760). This kit incorporates dUTP during second-strand cDNA synthesis and therefore only the cDNA strand generated during first-strand synthesis is sequenced. Adapter-ligated library was completed by PCR with Illumina PE primers. The resulting purified cDNA library was applied to an Illumina flow cell for cluster generation and sequenced on an Illumina sequencer (HiSeq Sequencing v4 Chemistry). Single-end sequenced reads were analyzed with the next*presso* pipeline[70]. Briefly, sequencing quality was checked with FastQC v0.11.7 (http://www.bioinformatics.babraham.ac.uk/projects/fastqc/), reads were aligned to the mouse reference genome (NCBI37/mm9, https://ccb.jhu.edu/software/tophat/igenomes.shtml) with TopHat-2.0.10 (ref. [71]) using Bowtie 1.0.0 (ref. [72]) and Samtools 0.1.19 (ref. [73]), allowing two mismatches and 20 multihits. Read counts were obtained with HTSeq-count v0.6.1 (ref. [74]), using the mouse NCBI37/mm9 gene annotation from https://ccb.jhu.edu/software/tophat/igenomes.shtml. Differential expression was performed with DESeq2 (ref. [75]), using a 0.05 FDR. GSEAPreranked[76] was used to perform gene set enrichment analysis for several gene signatures on a pre-ranked gene list, setting 1000 gene set permutations. Only those gene sets with significant enrichment levels (FDR $q$-value <0.25) were considered.

**Reporting summary.** Further information on research design is available in the Nature Research Reporting Summary linked to this article.

## Data availability
Source data for Figs. 1–4, 7 and Supplementary Figs. 1–7 are provided with the paper. RNAseq data have been deposited in GEO database (GSE127475). All other data are available from the corresponding author upon reasonable request.

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

## Acknowledgements

We thank Sara Kozma for kindly sharing the *S6K1*−/− mouse. We thank A. Efeyan and A. Ortega for helpful discussions. We are indebted to D. Megias for confocal microscopy analysis. Research in the Blasco Lab is funded by the Spanish Ministry of Economy and Competitiveness Projects (SAF2013-45111-R and SAF2015-72455-EXP), the Comunidad de Madrid Project (S2017/BMD-3770), the World Cancer Research (WCR) Project (16-1177), and the Fundación Botín (Spain).

## Author contributions

M.A.B. had the original idea and secured funding. M.A.B. and P.M. supervised research. M.A.B., P.M., and I.F.R. wrote the paper. I.F.R. and P.M. analyzed the data and performed experiments. S.S., K.W., and L.T.P. assisted with experiments. O.G. performed the bioinformatic analysis. R.S. was responsible for animal maintenance and assisted with animal experimentation. E.H. and C.B.A. analyzed rapamycin levels in plasma and liver samples. J.M.F. performed pathological analysis.

## Competing interests

The authors declare no competing interests.
