## [Peer Review File · Nature Communications]

Reviewers' comments:

Reviewer #2 (Remarks to the Author):

This is an interesting and potentially important paper from the laboratory of Maria Blasco, a leader in the investigation of how telomeres affect the lifespan and health of mice. Here, Blasco and colleagues report that inhibition of mTORC1 signaling by rapamycin or by deletion of S6K1 shortens the lifespan of telomerase deficient mice. If true, this is important from both a biology perspective and the translational treatment of progeroid diseases. However, the manuscript suffers from a significant failure to properly analyze (with correct statistical tests) and discuss their longevity results as well as other data, casting doubt on these findings. There is also some question surrounding rapamycin pharmacology and signaling in non-heaptic tissues that should be addressed. If the authors can satisfactorily address these issues, and if the results remain similar following correct statistical analysis, the results presented here will be a major addition to the literature regarding rapamycin, telomere biology, and mTOR signaling, and serve as a major advance in these fields.

1. Fundamentally, a major issue is one of multiple comparisons - the log-rank test utilized to compare lifespans is not corrected for the many comparisons conducted. This is evident in figures 1b/1c but is not critical given the high statistical significance of findings - but the p-values should nonetheless be corrected for the 4 comparisons tests. Note: these statements also apply to figure S1 based on the same data, which also needs to be corrected.

2. However, the results of Figure 6 are HIGHLY likely to be affected. In particular, the authors report 2 results where $0.01 < p < 0.05$; and they have made 4 comparisons. As such, it is extremely likely that one of both of these results may not be significant, especially given the small "n", and it is imperative that these results be properly analyzed.

3. A related issue is that the discussion fails to note that S6K1 deletion negatively impacts the lifespan ONLY of G3 Terc mice, not G1 or G2. In contrast, Figure 1 reports that rapamycin impairs the lifespan of G2 Terc mice. The reason for this discrepancy is not clear.

4. While the authors properly checked telomere length in rapamycin treated animals, it is imperative to also check the S6K1 Terc G2 and G3 mice for telomere length, as a trivial explanation for the results presented in Figure 6 is that deletion of S6K1 hastens telomere loss. I feel this is particularly important as the effect of S6K1 deletion on lifespan - if it is statistically significant after correction - only occurs in the G3s.

4. While the liver is undoubtedly important in the regulation of lifespan, the authors should examine pS6 signaling in Terc-deficient mice in other tissues. mTORC1 activity in heart, skeletal muscle, and other tissues have all been shown to impact longevity and healthspan.

5. A trivial explanation for the failure of chronic rapamycin treatment to suppress pS6 signaling in Terc G2 livers is that xenobiotic response pathways have been upregulated resulting in degradation of rapamycin in the liver. Since such an effect has never been previously observed, it is absolutely imperative that before drawing the conclusion that Terc G2 mice have compensatory increased mTORC1 signaling that the authors check this possibility by measuring rapamycin levels in liver and, ideally, blood. The San Antonio Nathan Shock Center Pharmacology Core provides rapamycin measurement in mouse tissues and blood on a fee-for-service basis.

6. MAJOR issue: t-tests are used throughout the manuscript where an ANOVA followed by post-tests which correct for multiple comparisons are more appropriate - in any case, SOME correction for multiple corrections must be performed. See fig 1D-1G, 2A-B, 3A, 3C, 3F, 3H, 3I, and Figure S3-S4.

Reviewer #3 (Remarks to the Author):

This manuscript shows that second generation Terc-/- have reduced lifespan when treated with

rapamycin. Additionally, that deletion of S6K1 decreases lifespan in the context of telomerase deficiency.

This is a very interesting observation and at first glance puzzling- particularly since rapamycin as well as deletion of S6K1 has been shown to extend lifespan of wild-type mice. Mechanistically, it is not clear why the opposite is happening in TERC^{-/-} mice. One possibility would be that rapamycin affects cancer progression differently in Terc^{-/-} which have been shown to be cancer resistant to a certain degree. I think that this could be a reasonable explanation - the effects of rapamycin in extending lifespan in wild-type mice are mainly due to tumor suppression as suggested by others, however, these beneficial effects are not observed in Terc^{-/-} mice which are cancer resistant. The paper is relatively preliminary in terms of the molecular characterization of the phenotypes- the analysis of the downstream pathways activated by telomere dysfunction which impact on regenerative capacity such as apoptosis, senescence, proliferation has not been conducted. mTOR has been shown to impact on several metabolic pathways such as mitochondrial biogenesis/mitophagy. Since late generation TERC^{-/-} have been shown to have decreased mitochondrial function- could it be that rapamycin is merely enhancing this phenotype to levels which are incompatible with mouse survival? Authors claim that telomere dysfunction causes increased mTOR activity and metabolism- which is inconsistent with previous data.

A few general comments:

The gender effects of rapamycin on lifespan extension in wild-type mice are puzzling considering several published reports showing stronger effects in females than in males in genetically heterogeneous mice. According to Miller et al. 2014 the gender effects occur regardless of the dose used.

Authors used a relatively high dose of rapamycin. Could it be that this is a question of dose and that a lower dose would impact differently on lifespan? It has been suggested previously that during aging, mice with dysfunctional telomeres develop weight loss and show a phenotype similar to starvation characterized by glucose depletion and in fact reduced mTOR activity (Missios 2014)- authors should discuss difference between both studies. Could it be that effects of rapamycin on glucose metabolism or glucose absorption in the intestine could be affecting survival?

Aging phenotypes are diverse and I believe the manuscript would benefit from a much more detailed analysis of aging phenotypes (eg. Frailty, exercise capacity, musculoskeletal function, glucose tolerance test and many others). A lot of emphasis has been placed on lifespan- but equally important is a characterization of the healthspan of the mice.

The median lifespan of the wild-type mice are relatively low compared to other published studies and my own experience in aging this mouse strain. We usually observe a median lifespan of wild-type mice around 28-29 months in C57 BL6. Facilities and mouse maintenance could explain inter-laboratory differences and should be discussed.

The authors measured FISH telomere fluorescence however did not measure critically short telomeres and dysfunctional telomeres (TIF). I am surprised that the differences in maximum lifespan are so acute only after 2 generations. The maximum lifespan of 2nd generation Terc^{-/-} seems to vary considerably between labs. Even in previous publications from the Blasco lab maximum lifespan of 2nd generation Terc^{-/-} mice in the same strain were higher. Authors did not find any effects of rapamycin on telomere length- authors should consider what happens to frequencies of TIF- particularly since previous work showed rapamycin reduces TIF in mice (Correia-Melo 2017).

The beneficial effects of rapamycin in terms of intestinal atrophy in G2 Terc^{-/-} are interesting. I wonder how this impacts on stem and progenitor cell function and proliferation in the intestine (a more careful characterization could be conducted in my view).

In Figure 3D I can clearly see that one of the bands (in Rapamycin treated TERC^{-/-} male mice) has been cropped. For these sort of analyses I would advise authors to present western blots of at least 3 animals per group in the same blot.

Rapamycin has been shown to suppress inflammaging- is this affected?

The data on the S6K1 mice supports the previous findings with rapamycin- however the characterization of the mice phenotypically and molecularly is non-existent.

Detailed Answer to Reviewers' comments:

NOTE: We did not receive the commentaries by Reviewer #1

Detailed Answers to Reviewer #2:

This is an interesting and potentially important paper from the laboratory of Maria Blasco, a leader in the investigation of how telomeres affect the lifespan and health of mice. Here, Blasco and colleagues report that inhibition of mTORC1 signaling by rapamycin or by deletion of S6K1 shortens the lifespan of telomerase deficient mice. If true, this is important from both a biology perspective and the translational treatment of progeroid diseases. However, the manuscript suffers from a significant failure to properly analyze (with correct statistical tests) and discuss their longevity results as well as other data, casting doubt on these findings. There is also some question surrounding rapamycin pharmacology and signaling in non-hepatic tissues that should be addressed. If the authors can satisfactorily address these issues, and if the results remain similar following correct statistical analysis, the results presented here will be a major addition to the literature regarding rapamycin, telomere biology, and mTOR signaling, and serve as a major advance in these fields.

ANSWER: First of all, we would like to kindly thank the reviewer for considering that “**This is an interesting and potentially important paper from the laboratory of Maria Blasco, a leader in the investigation of how telomeres affect the lifespan and health of mice**”. The reviewer also has a number of concerns which we can address in full in a revised manuscript as follows:

1. Fundamentally, a major issue is one of multiple comparisons - the log-rank test utilized to compare lifespans is not corrected for the many comparisons conducted. This is evident in figures 1b/1c but is not critical given the high statistical significance of findings - but the p-values should nonetheless be corrected for the 4 comparisons tests. Note: these statements also apply to figure S1 based on the same data, which also needs to be corrected.

ANSWER: We are comparing the data pairwise, namely the difference between rapamycin treatment versus untreated mice within the same genotype. We have now removed from the figures the comparisons between mice of different genotypes (see **new Fig. 1** and **new Supplementary Fig. 1**). Thus, we are not performing multiple comparisons as the reviewer points out. We indicated the exact p-values in the relevant comparisons. We would like to highlight that the log-rank test used here are commonly used in many other papers determining mouse survival from our lab and other labs¹⁻⁵. We could represent the data corresponding to the same genotype in different plots but for space constriction we rather leave these figures as they are now.

2. However, the results of Figure 6 are HIGHLY likely to be affected. In particular, the authors report 2 results where $0.01 < p < 0.05$; and they have made 4 comparisons. As such, it is extremely likely that one of both of these results may not be significant, especially given the small "n", and it is imperative that these results be properly analyzed.

ANSWER: As mentioned above, we are comparing the data pairwise, namely the difference between S6K1-deficient versus S6K1-proficient mice within the same *Terc* generation group (*Terc*^{+/+}, G1 *Terc*^{-/-}, G2 *Terc*^{-/-} and G3 *Terc*^{-/-}) (see **new Fig. 7**). Thus, we are not

performing multiple comparisons as the reviewer points out. We indicated the exact p-values in the relevant comparisons. We could represent the data corresponding to the same telomerase generation in different plots but for space constriction we rather leave these figures as they are now.

3. A related issue is that the discussion fails to note that S6K1 deletion negatively impacts the lifespan ONLY of G3 Terc mice, not G1 or G2. In contrast, Figure 1 reports that rapamycin impairs the lifespan of G2 Terc mice. The reason for this discrepancy is not clear.

ANSWER: We discussed this issue in the revised manuscript text. *“While chronic rapamycin affects negatively already in G2 Terc^{-/-}, the effects of S6K1 deficiency become apparent in G3. This discrepancy might be explained by different telomere length due to different genetic backgrounds of both mouse colonies and by the fact that S6K1 and S6K2 present functional compensation and redundancy”* (page 27, 2nd paragraph).

4. While the authors properly checked telomere length in rapamycin treated animals, it is imperative to also check the S6K1 Terc G2 and G3 mice for telomere length, as a trivial explanation for the results presented in Figure 6 is that deletion of S6K1 hastens telomere loss. I feel this is particularly important as the effect of S6K1 deletion on lifespan - if it is statistically significant after correction - only occurs in the G3s.

ANSWER: In the revised manuscript, we have included QFISH analysis of both liver and intestine of the different mutant mice (**new Fig. S6**). These results clearly show that S6K1 deficiency does not affect telomere shortening in Terc^{-/-} mice.

4. While the liver is undoubtedly important in the regulation of lifespan, the authors should examine pS6 signaling in Terc-deficient mice in other tissues. mTORC1 activity in heart, skeletal muscle, and other tissues have all been shown to impact longevity and healthspan.

ANSWER: In the revised manuscript, we have included data of pS6 levels in skeletal muscle and in heart (**see new Fig. 3F-I**). As observed in liver samples, the new results show that rapamycin treatment significantly reduces pS6 levels in skeletal muscle and heart from wild-type mice but not from G2 Terc^{-/-} mice.

5. A trivial explanation for the failure of chronic rapamycin treatment to suppress pS6 signaling in Terc G2 livers is that xenobiotic response pathways have been upregulated resulting in degradation of rapamycin in the liver. Since such an effect has never been previously observed, it is absolutely imperative that before drawing the conclusion that Terc G2 mice have compensatory increased mTORC1 signaling that the authors check this possibility by measuring rapamycin levels in liver and, ideally, blood. The San Antonio Nathan Shock Center Pharmacology Core provides rapamycin measurement in mouse tissues and blood on a fee-for-service basis.

ANSWER: We thank the reviewer for this suggestion. In the revised manuscript we have analyzed the rapamycin levels in liver and blood samples corresponding to wild type and G2 Terc^{-/-} mice (**new Fig. S2A,B**). The results show no significant differences in rapamycin levels between wildtype and G2 Terc^{-/-} livers of both genders. The rapamycin levels in fasted and fed male mice of both genotypes did not reveal either significant differences between genotypes.

6. MAJOR issue: t-tests are used throughout the manuscript where an ANOVA followed by post-tests which correct for multiple comparisons are more appropriate - in any case, SOME correction for multiple corrections must be performed. See fig 1D-1G, 2A-B, 3A, 3C, 3F, 3H, 3I, and Figure S3-S4.

ANSWER: In the revised manuscript we have applied the One-way ANOVA followed by Tukey post-test as suggested by the reviewer.

Detailed Answers to Reviewer #3:

This manuscript shows that second generation *Terc*^{-/-} have reduced lifespan when treated with rapamycin. Additionally, that deletion of S6K1 decreases lifespan in the context of telomerase deficiency. This is a very interesting observation and at first glance puzzling-particularly since rapamycin as well as deletion of S6K1 has been shown to extend lifespan of wild-type mice. Mechanistically, it is not clear why the opposite is happening in *Terc*^{-/-} mice. One possibility would be that rapamycin affects cancer progression differently in *Terc*^{-/-} which have been shown to be cancer resistant to a certain degree. I think that this could be a reasonable explanation - the effects of rapamycin in extending lifespan in wild-type mice are mainly due to tumor suppression as suggested by others, however, these beneficial effects are not observed in *Terc*^{-/-} mice which are cancer resistant.

ANSWER: First, we would like to thank the reviewer for considering the interest and novelty of our observations. In particular, that ***"This is a very interesting observation and at first glance puzzling- particularly since rapamycin as well as deletion of S6K1 has been shown to extend lifespan of wild-type mice"***.

Regarding cancer, we clearly show in the manuscript that rapamycin has a cancer-independent effect on prolonging survival (see **Fig. 1C** and **Supplementary Fig. 1 C,D**). In addition, full histopathological analysis of G2 *Terc*^{-/-} mice at the human end-point shows that rapamycin did not abolish the cancer-suppressor effect of telomerase deficiency (**Fig 1 F-G**).

The paper is relatively preliminary in terms of the molecular characterization of the phenotypes- the analysis of the downstream pathways activated by telomere dysfunction which impact on regenerative capacity such as apoptosis, senescence, proliferation has not been conducted.

ANSWER: We respectfully disagree with the reviewer that our findings are preliminary, as we have showed most of these analyses requested by the reviewer in the manuscript. We think the reviewer must have missed these data in the manuscript. In particular, in **Supplementary Fig. 3A-E**, we already showed the analysis of cells positive for γ H2AX, p53, p21, apoptosis (AC3), and telomere-induced damage foci (TIFs) at the human end point. In the revised manuscript, we also analyzed γ H2AX, p53, p19 and pH3 in mice treated with rapamycin during 2 month (see **new Fig. S3F,I**). We observe that rapamycin does not affect the downstream pathways activated by telomere dysfunction since we did not find significant differences in the number of positive cells for γ H2AX, p53, p21, p19, AC3, pH3 between untreated and rapamycin-treated mice of both genotypes.

mTOR has been shown to impact on several metabolic pathways such as mitochondrial biogenesis/mitophagy. Since late generation *Terc*^{-/-} have been shown to have decreased mitochondrial function- could it be that rapamycin is merely enhancing this phenotype to levels which are incompatible with mouse survival? Authors claim that telomere dysfunction causes increased mTOR activity and metabolism- which is inconsistent with previous data.

ANSWER: We have measured mitochondrial DNA copy number by q-PCR in the liver (see **new Fig. 4H**). *“In agreement with previous work ⁶, untreated G2 Terc^{-/-} mice presented reduced mtDNA copy number as compared to untreated wildtype indicating worsened mitochondrial biogenesis in telomerase deficient mice (Fig. 4H). Rapamycin treatment lead to a decrease in the mtDNA copy number in wildtype mice while no effects were seen in G2 Terc^{-/-} mice.”* (page 17, 2nd paragraph). We have also measured the levels of autophagy by analyzing p62 levels by WB (see **new Fig. S5A**). We did not observe any differences between wt and G2 Terc^{-/-} mice regardless of the treatment.

A few general comments: The gender effects of rapamycin on lifespan extension in wild-type mice are puzzling considering several published reports showing stronger effects in females than in males in genetically heterogeneous mice. According to Miller et al. 2014 the gender effects occur regardless of the dose used. Authors used a relatively high dose of rapamycin. Could it be that this is a question of dose and that a lower dose would impact differently on lifespan? It has been suggested previously that during aging, mice with dysfunctional telomeres develop weight loss and show a phenotype similar to starvation characterized by glucose depletion and in fact reduced mTOR activity (Missios 2014)- authors should discuss difference between both studies. Could it be that effects of rapamycin on glucose metabolism or glucose absorption in the intestine could be affecting survival? Aging phenotypes are diverse and I believe the manuscript would benefit from a much more detailed analysis of aging phenotypes (eg. Frailty, exercise capacity, musculoskeletal function, glucose tolerance test and many others).

ANSWER: We have performed several metabolic analyses including glucose tolerance test (GTT), Insulin tolerance test (ITT) quantification of fasting IGF-1, insulin and glucose levels as well as the insulin resistance HOMA-IR index (see **new Fig. 4A-G**), but in this case we did not see differences. The results are described as follows *“To address the rapamycin effects on the response to glucose and insulin we performed a glucose (GTT) and insulin (ITT) tolerance tests on these mouse cohorts. We found that wildtype mice treated with rapamycin are more glucose intolerant compared to untreated wildtype mice, in accordance to previous work ^{4,7}. However, rapamycin treatment did not alter the glucose response in G2 Terc^{-/-} (Fig. 4A). In particular, the area-under-the-curve (AUC) values for the GTT assays were significantly higher in wildtype mice treated with rapamycin as compared with controls (Fig. 4B). Rapamycin treatment however did not affect the response to exogenously administered insulin in neither of the genotypes (Fig. 4C). We measured the fasting levels of insulin, IGF1 and glucose. As already reported ⁴, rapamycin did not lead to significant alterations in either fasting insulin or fasting IGF1 but led to higher fasting glucose levels in both genotypes (Fig. 4D-F). We calculated the insulin resistance HOMA-IR index and the results revealed a worsened insulin resistance in both wild-type and G2 Terc^{-/-} mice treated with rapamycin (Fig. 4G)”* (page 17, 1st paragraph).

We agree with the reviewer that neuromuscular and cognitive tests as a measurement of other aging phenotypes that could potentially be affected by rapamycin treatment are of interest and deserve future investigation. However, we think that these studies are out of the scope of this work. In addition, due to the fact that all the mice included in this study are already dead makes it impossible to performed these tests.

In the revised manuscript, we have included a discussion about the discrepancy between our work and Miller's ⁴ regarding the gender effects of rapamycin on lifespan extension in wild-type mice. *“Of interest, wild-type mice treated with rapamycin showed an increase*

that is higher than that reported by other authors using the same diet in genetically heterogenous mice⁴. In contrast to our results showing that chronic rapamycin treatment led to higher increased in lifespan in males (43%) than in females (23%), Miller et al. reported that rapamycin increased median lifespan to a greater extent in females (26%) than in males (23%)⁴. These discrepancies may be due to different genetic backgrounds and to the fact that we initiated the rapamycin treatment at three months of age instead of nine month of age⁴, suggesting that earlier intervention may have greater effects” (page 25, 1st paragraph).

A lot of emphasis has been placed on lifespan- but equally important is a characterization of the healthspan of the mice.

ANSWER: As indicated above, we have included the new metabolic data mentioned above. Nevertheless, we would like to point out that the survival curves show that rapamycin treatment lengthens the time period in which all the mice are still alive.

The median lifespan of the wild-type mice is relatively low compared to other published studies and my own experience in aging this mouse strain. We usually observe a median lifespan of wild-type mice around 28-29 months in C57 BL6. Facilities and mouse maintenance could explain inter-laboratory differences and should be discussed. The authors measured FISH telomere fluorescence however did not measured critically short telomeres and dysfunctional telomeres (TIF). I am surprised that the differences in maximum lifespan are so acute only after 2 generations. The maximum lifespan of 2nd generation *Terc*^{-/-} seems to vary considerably between labs. Even in previous publications from the Blasco lab maximum lifespan of 2nd generation *Terc*^{-/-} mice in the same strain were higher. Authors did not find any effects of rapamycin on telomere length- authors should consider what happens to frequencies of TIF- particularly since previous work showed rapamycin reduces TIF in mice (Correia-Melo 2017).

ANSWER: Again, the reviewer must have missed this data in the manuscript, we indeed quantified telomere damage (TIFs) in intestinal sections and found that rapamycin treatment does not alter the number of cells showing TIFs as compared to untreated mice of both genotypes (see **Supplementary Fig. 3E**). Our results are indeed in agreement with Correia-Melo et al.⁸. They observed that rapamycin treatment did not alter the incidence of TIFs positive cells in the lung or in the liver of wildtype mice. They did, however, observe an increase in TIFs frequency in mice deficient for *nf- κ b1*, a mouse model for enforced inflammation, that was rescued by rapamycin. Our genetic model, namely the telomerase deficient mice is a model of premature aging but not of chronic inflammation.

In the revised manuscript, we have discussed the potential reasons underlying the different lifespan observed among different labs with mice of the same genotype. *“inter-laboratory differences in median lifespan might be attributed to housing conditions and to continuous inbreeding for mouse colony maintenance”* (Page 25, 1st paragraph).

The beneficial effects of rapamycin in terms of intestinal atrophy in G2 *Terc*^{-/-} are interesting. I wonder how this impacts on stem and progenitor cell function and proliferation in the intestine (a more careful characterization could be conducted in my view).

ANSWER: We think that the apparent beneficial effects of rapamycin in G2 *Terc*^{-/-} mice on the severity of intestinal pathologies at death are due to the difference in life-span between untreated and treated mice (**Fig. 1H,I** and **Fig. S2C,D**). Indeed, when we quantified telomere damage (TIFs) and the number of mitotic cells in intestinal sections we found that rapamycin treatment neither alter the number of cells showing TIFs nor the number of pH3 positive cells as compared to untreated mice of both genotypes (**Fig. S3E,I**). We discuss this in the text “*We found that 100% of G2 Terc^{-/-} mice presented intestinal atrophy at their endpoint in contrast to wild-type cohorts that did not present intestinal atrophy independently of the diet (Fig. 1H). Rapamycin fed G2 Terc^{-/-} mice showed less severe intestinal atrophies compared to the control diet counterparts (Fig. 1H). In particular, while 50% of control diet G2 Terc^{-/-} mice presented severe intestinal atrophy, 80% of rapamycin fed G2 Terc^{-/-} mice showed mild atrophy (Fig. 1H). Although these findings may suggest that rapamycin treatment ameliorates intestinal atrophy in telomerase-deficient mice, the fact that rapamycin treated G2 Terc^{-/-} mice died at an earlier timepoint (2 months earlier) compared to the control diet counterparts (Fig. 1I) may also explain the lower severity of intestinal atrophy. This notion is supported by the fact that, when we separated mice by gender, only rapamycin treated G2 Terc^{-/-} males but not females showed significantly decreased severe intestinal lesions (Fig. S2C,D), in agreement with the fact that only rapamycin treated males showed a significantly decreased longevity compared to G2 Terc^{-/-} controls (see Fig. 1I; Fig. S2C,D)*” (page 11, 1st paragraph).

In Figure 3D I can clear see that one of the bands (in Rapamycin treated TERC^{-/-} male mice) has been cropped. For these sorts of analyses I would advise authors to present western blots of at least 3 animals per group in the same blot.

ANSWER: We cropped samples 4 and 5 corresponding to RAPA treated wild type mice because the WB for total S6 had a transfer bubble in these lanes. We include below the original WB images of new Fig. S5B as well as a loading control (SMC1) for the reviewer. In Fig. S5B we clearly mark with a vertical line where the images have been cropped. We also include a new WB with three animals per group showing pS6 and tubuline as the loading control.

Rapamycin has been shown to suppress inflammaging- is this affected? The data on the S6K1 mice supports the previous findings with rapamycin- however the characterization of the mice phenotypically and molecularly is non existent.

ANSWER: In the reviewed manuscript, we have included pS6 and telomere length analysis of S6K1 *Terc* mouse cohorts (new Fig. S6).

References

1. Tomás-Loba, A. *et al.* Telomerase Reverse Transcriptase Delays Aging in Cancer-Resistant Mice. *Cell* **135**, 609–622 (2008).
2. Selman, C. *et al.* Ribosomal protein S6 kinase 1 signaling regulates mammalian life span. *Science* (80-.). (2009). doi:10.1126/science.1177221
3. Harrison, D. E. *et al.* Rapamycin fed late in life extends lifespan in genetically heterogeneous mice HHS Public Access. *Nature* (2009). doi:10.1038/nature08221
4. Miller, R. A. *et al.* Rapamycin-mediated lifespan increase in mice is dose and sex dependent and metabolically distinct from dietary restriction. *Aging Cell* (2014). doi:10.1111/accel.12194
5. Martínez, P., Gómez-López, G., Pisano, D. G., Flores, J. M. & Blasco, M. A. A genetic interaction between RAP1 and telomerase reveals an unanticipated role for

- RAP1 in telomere maintenance. *Aging Cell* **15**, 1113–1125 (2016).
6. Missios, P. *et al.* Glucose substitution prolongs maintenance of energy homeostasis and lifespan of telomere dysfunctional mice. *Nat. Commun.* (2014). doi:10.1038/ncomms5924
 7. Lamming, D. W. *et al.* Rapamycin-induced insulin resistance is mediated by mTORC2 loss and uncoupled from longevity. *Science* (80-.). (2012). doi:10.1126/science.1215135
 8. Correia-Melo, C. *et al.* Rapamycin improves healthspan but not inflammaging in *nfkb1* $-/-$ mice. *Aging Cell* (2019). doi:10.1111/accel.12882
 9. Brust, V., Schindler, P. M. & Lewejohann, L. Lifetime development of behavioural phenotype in the house mouse (*Mus musculus*). *Frontiers in Zoology* (2015). doi:10.1186/1742-9994-12-S1-S17

Reviewers' comments:

Reviewer #2 (Remarks to the Author):

As noted, this is an interesting and potentially important paper from the laboratory of Maria Blasco, a leader in the investigation of how telomeres affect the lifespan and health of mice. The authors have done an excellent job of responding to the comments of all the reviewers, and with one exception regarding the statistics surrounding figure 7, and a few minor issues, this powerful manuscript will be a terrific addition to the literature surrounding aging, mTOR, and rapamycin.

Major:

Multiple comparisons:

Multiple comparison issues in most panels throughout the manuscript have now been satisfactorily resolved.

However, with regard to Figure 7 - fundamentally, if many statistical tests are performed, the odds are good that one will have a $p < 0.05$ (e.g., <https://xkcd.com/882/>). As such, Figure panel 7C has a fundamental issue - 4 tests have been performed. The fact that they are log-rank tests rather than t-tests is irrelevant - a p value of 0.033 is not significant following Bonferroni correction... and using a Holm-Sidak correction here provides a two-sided p value of only 0.126 instead of 0.033.

One potential issue is that the authors are using a log-rank test - which assumes a constant hazard rate - may be inappropriate as the curves slowly converge, suggesting that an assumption of proportional hazards which underlies that log-rank test is not appropriate. A Wilcoxon test, which assumes a difference in hazard rate, but does not assume proportional hazards, might therefore be a more appropriate test for the G3 mice. Another possibility is that the Fig 7A/B G3 mice could be analyzed by cox regression, with sex and genotype as factors, rather than somewhat messily pooling sexes.

I would probably conclude that the authors consider examining Figure 7A, B, and C (G3) using Wilcoxon rather than log-rank; and if not significant, then I would list the multiple comparisons as a limitation in the discussion of the conclusions arising from Figure 7C.

Minor:

Figure 1 legend, Figure 6 C, and throughout: The authors use "gender" where they mean "sex", mice do not have gender which is a social construct.

Figure 1: Male/female is not distinguished or labeled in figure panels B and C or in the legend.

Reviewer #3 (Remarks to the Author):

I am thankful that the authors attempted to answer so many of the questions posed by the reviewers. I find the observations very interesting and potentially important. However, the aspect that limits my enthusiasm is that there is not a clear mechanistic explanation for their findings. However, given the multitude of pathways affected by the mTOR complex- unraveling this could be a huge undertaking.

I have some issues with some of the new experiments:

I appreciate the fact that authors explored the potential impact of rapamycin on mtDNA copy number by q-PCR as a potential mechanism. However, in order to rule out the involvement of mitochondria, more comprehensive analysis of mitochondrial function should be conducted. Similarly, measurements of p62 levels do not per se indicate the functionality of autophagy following rapamycin treatment.

Figure 4 attempts to address the impact of rapamycin in glucose metabolism by measuring glucose and insulin tolerance however, in majority graphs of there are groups with n=2 animals- how can statistical analysis be performed or any conclusions be obtained with n=2? Authors claim one way ANOVA was performed. These data should be seen as preliminary.

Authors analyzed the intestine of wild-type and G2TERC^{-/-} mice for markers of DDR and senescence and found that rapamycin did not impact on those (it was not clear to me which ages the mice were analyzed). Work by the Campisi lab showed that rapamycin did not impact on senescence cell-cycle arrest mediated pathways but specifically on the SASP (Laberge 2015) and that this could explain its beneficial effects. Could other tissues be analyzed? Also not clear which areas/cells in the intestine were analyzed?

Detailed Answer to Reviewers' comments:

Detailed Answers to Reviewer #2:

As noted, this is an interesting and potentially important paper from the laboratory of Maria Blasco, a leader in the investigation of how telomeres affect the lifespan and health of mice. The authors have done an excellent job of responding to the comments of all the reviewers, and with one exception regarding the statistics surrounding figure 7, and a few minor issues, this powerful manuscript will be a terrific addition to the literature surrounding aging, mTOR, and rapamycin.

ANSWER: We appreciate reviewer's consideration of our work as an "interesting and potentially important paper" and a "powerful manuscript that will be a terrific addition to the literature surrounding aging, mTOR, and rapamycin". The reviewer also has a number of concerns which we have addressed in full in a revised manuscript as described below:

1. Multiple comparison issues in most panels throughout the manuscript have now been satisfactorily resolved. However, with regard to Figure 7 - fundamentally, if many statistical tests are performed, the odds are good that one will have a $p < 0.05$ (e.g., <https://xkcd.com/882/>). As such, Figure panel 7C has a fundamental issue - 4 tests have been performed. The fact that they are log-rank tests rather than t-tests is irrelevant - a p value of 0.033 is not significant following Bonferroni correction... and using a Holm-Sidak correction here provides a two-sided p value of only 0.126 instead of 0.033.

One potential issue is that the authors are using a log-rank test - which assumes a constant hazard rate - may be inappropriate as the curves slowly converge, suggesting that an assumption of proportional hazards which underlies that log-rank test is not appropriate. A Wilcoxon test, which assumes a difference in hazard rate, but does not assume proportional hazards, might therefore be a more appropriate test for the G3 mice. Another possibility is that the Fig 7A/B G3 mice could be analyzed by cox regression, with sex and genotype as factors, rather than somewhat messily pooling sexes.

I would probably conclude that the authors consider examining Figure 7A, B, and C (G3) using Wilcoxon rather than log-rank; and if not significant, then I would list the multiple comparisons as a limitation in the discussion of the conclusions arising from Figure 7C.

ANSWER: We thank the reviewer for his/her explanation about potential misinterpretations of the survival results by the use of the log-rank test which assumes a constant hazard rate and gives equal weight to all the time points. In contrast, the Wilcoxon test does not assume proportional hazards and gives more weight to deaths at early time points and might therefore be a more appropriate test for late generation of telomerase knock-out mice (G2-G3). In the **new Figure 7A-C**, we now include the p-values from both statistical tests, log-rank and Wilcoxon. Indeed, the p-values of late generation Terc-ko (G3) becomes more significant with the Wilcoxon test than with the Log-rank whereas the p-values of the Terc-WT cohorts are more significant with the Log-rank test. These results clearly reinforce our statement that G3 *S6K1^{-/-} Terc^{-/-}* mice (females and both sexes combined) present a reduced median survival as compared to G3 *S6K1^{+/+} Terc^{-/-}*. In the revised manuscript, we have included a full histopathological analysis of *S6K1 Terc* mouse cohorts at HEP showing the incidence of cancer (lymphomas and sarcomas) and

intestine degenerative pathologies, supporting that mTORC1 inhibition exerts a protective role in lymphoma development and does not affect on short telomeres-induced intestinal atrophies (**New Supplementary Figure 7D-F; see page 23-24, lines 505-514**).

2. Figure 1 legend, Figure 6 C, and throughout: The authors use "gender" where they mean "sex", mice do not have gender which is a social construct.

ANSWER: We have replaced gender by sex.

3. Figure 1: Male/female is not distinguished or labeled in figure panels B and C or in the legend.

ANSWER: We have indicated the sexes in panel B and C of Figure 1.

Detailed Answers to Reviewer #3:

I am thankful that the authors attempted to answer so many of the questions posed by the reviewers. I find the observations very interesting and potentially important. However, the aspect that limits my enthusiasm is that there is not a clear mechanistic explanation for their findings. However, given the multitude of pathways affected by the mTOR complex-unraveling this could be a huge undertaking.

ANSWER: First, we would like to thank the reviewer for appreciating our efforts to answer his/her questions and for considering our results “**very interesting and potentially important**”. We also appreciate his opinion that fully unraveling the molecular mechanisms underlying our findings might not be possible given the plethora of pathways affected by the mTOR complex. The reviewer also has a number of concerns which we have addressed in full in a revised manuscript as described below:

1. I appreciate the fact that authors explored the potential impact of rapamycin on mtDNA copy number by q-PCR as a potential mechanism. However, in order to rule out the involvement of mitochondria, more comprehensive analysis of mitochondrial function should be conducted.

ANSWER: To expand the potential impact of rapamycin on mtDNA, we have also addressed the mtDNA copy number in muscle (**new Figure 4C**). The results confirm that rapamycin decreases mitochondrial content in wild-type mice and that G2 *Terc*^{-/-} show lower number of mitochondria as compared to wild-type mice, both in liver and muscle. Rapamycin did not further lower mitochondrial content in G2 *Terc*^{-/-}. In addition, in the revised manuscript, we further demonstrate the decreased mitochondria content in rapamycin-treated wild-type and G2 *Terc*^{-/-} mice by analyzing by western blot the levels of ATP5A, UQCRC2, MTCO1 and SDHB, components of mitochondrial complex V, III, IV and II, respectively (**new Figure 4D**). These data support the lack of effect of rapamycin on mitochondria function in G2 *Terc*^{-/-} mice, ruling out the involvement of mitochondria in the rapamycin-mediated effect on the survival of telomerase deficient mice (see **page 18, line 377-386**).

2. Measurements of p62 levels do not per se indicate the functionality of autophagy following rapamycin treatment.

ANSWER: We thank the reviewer for this comment. We have now analyzed the microtubule-associated protein 1A/1B-light chain 3 (LC3), the most widely monitored autophagy-related protein, in liver samples by western blot. During autophagy the cytosolic form of LC3 (LC3-I) is conjugated to phosphatidylethanolamine to form LC3-II that is then recruited to autophagosomes. LCII although larger in mass shows faster electrophoretic mobility in SDS-PAGE gels as a consequence of increased hydrophobicity (14 kDa of LC3-II as compared to 16 kDa of LC3-I). Thus, the ratio between LC3-II/LC3-I forms has become a reliable method for monitoring autophagy LC3-II marker (Klionsky et al., Autophagy 2016). The p62 western blots have been removed from the paper and replaced instead by the LC3 western blot results (**new Figure 4A**). As expected, mTORC1 inhibition by chronic rapamycin treatment induces autophagy in wild-type mice but does not alter autophagy levels in the G2 *Terc*^{-/-}

knock-out mice, again indicating a lack of effect of chronic rapamycin in telomerase-deficient mice with short telomeres (see page 17-18, lines 370-377).

3. Figure 4 attempts to address the impact of rapamycin in glucose metabolism by measuring glucose and insulin tolerance however, in majority graphs of there are groups with $n=2$ animals- how can statistical analysis be performed or any conclusions be obtained with $n=2$? Authors claim one way ANOVA was performed. These data should be seen as preliminary.

ANSWER: We understand reviewer's concern. When we performed these analyses there were very few remaining alive mice within the mouse cohorts under the study, limiting thereby the available mice to be included in the study of glucose metabolism. The fact the we confirmed previous published data showing that wild-type mice chronically treated with rapamycin developed glucose intolerance but the response to insulin remain unaffected (Lamming et al. 2012; Miller et al. 2014) made us confident in these data even dealing with a sample size of 2 in both the untreated and treated wild-type groups. Nevertheless, we have repeated the GTT and the ITT in an independent group of wild-type mice treated with rapamycin during two months. The results are shown in figure for reviewers below. We again observed an increase glucose intolerance in the rapamycin treated wild-type compared to control mice and no effect of the rapamycin treatment on the response to insulin. The fact that these new experiments are not performed in parallel with the *G2 Terc^{-/-}* mouse cohorts and that mice have been treated with rapamycin for two months (**Fig for reviewers**) instead of five months makes it impossible to be included in the paper figure. We therefore in the revised figures have removed the p values from the comparison between the untreated and rapamycin treated wild-type cohorts where $n=2$. We have moved these data to supplementary figures (**Supplementary Figure 5A-C**). In addition, we have replaced the data regarding fasting IGF1 levels for the data analysing the IGF1 levels in mice fed rapamycin for two months where the sample sizes have been considerably increased ($n=6-8$) (**new Supplementary Figure 5D**).

4. Authors analyzed the intestine of wild-type and G2TERC^{-/-} mice for markers of DDR and senescence and found that rapamycin did not impact on those (it was not clear to me which ages the mice were analyzed).

ANSWER: The age of the mice analyzed in Fig. S3 has been indicated in figure legend. *“The age of the wild type mice analyzed ranges between 20 and 30 months and the age of the G2 Terc^{-/-} between 6 and 7 months (A-E). The age of all mice analyzed from both genotypes ranges between 4 and 5 months (F-L)”*.

5. Work by the Campisi lab showed that rapamycin did not impact on senescence cell-cycle arrest mediated pathways but specifically on the SASP (Laberge 2015) and that this could explain its beneficial effects. Could other tissues be analyzed? Also not clear which areas/cells in the intestine were analyzed?

ANSWER: In the revised manuscript we included the analysis of γ H2AX, p53 and p21 levels in skeletal muscle (**new supplementary Figure 3J-L**). The results show that rapamycin does in fact not affect the DNA damage response elicited by shorten telomeres in a non-proliferative tissue such as the skeletal muscle. In addition, we have also analyzed the plasma IL6 levels in untreated and rapamycin treated wild-type and G2 Terc^{-/-} mice (**new supplementary Figure 4F**). In contrast to Campisi findings, we find that rapamycin does not affect the plasma IL6 levels neither in the wild-type nor in G2 Terc^{-/-} mice, ruling out that rapamycin-mediated life-span extension in wild-type mice is due to an effect on the SASP. The work by Campisi was performed in culture human cells while our work is performed in mice “in vivo”, which may explain the different findings (see **page 15, lines 315-322**). We have also studied the response to acute rapamycin in cultured G2 Terc^{-/-} Mefs (mouse embryonic fibroblasts) and found no differences as compared to wildtype Mefs with regards to mTORC1 inhibition (**Supplementary Figure 6**). The lack of effects of rapamycin in G2 Terc^{-/-} mice are shown in “in vivo” during chronic rapamycin treatment what is the major point of our work.

The IHQ analysis corresponding to molecular markers in intestine shown in Fig. S3 correspond to crypt cells. This has been indicated in Fig. S3 legend.

REVIEWERS' COMMENTS:

Reviewer #2 (Remarks to the Author):

As noted, this is an interesting and potentially important paper from the laboratory of Maria Blasco, a leader in the investigation of how telomeres affect the lifespan and health of mice. The authors have done an excellent job of responding to the comments of all the reviewers. In the opinion of this reviewer, the manuscript is now ready to be accepted.

A misspelling on line 76: "serin/threonin" should be "serine/threonine"

Reviewer #3 (Remarks to the Author):

Authors have responded adequately to my concerns. I recommend publication.

Detailed Answer to Reviewers' comments:

Detailed Answers to Reviewer #2:

Reviewer: As noted, this is an interesting and potentially important paper from the laboratory of Maria Blasco, a leader in the investigation of how telomeres affect the lifespan and health of mice. The authors have done an excellent job of responding to the comments of all the reviewers. In the opinion of this reviewer, the manuscript is now ready to be accepted.

Authors: We thank the reviewer for considering our work interesting and important and for acknowledging our job as excellent.

Reviewer: a misspelling on line 76: "serin/threonin" should be "serine/threonine"

Authors: The misspelling has been corrected

Detailed Answers to Reviewer #3:

Reviewer: authors have responded adequately to my concerns. I recommend publication.

Authors: We thank the reviewer for recommending our work for publication.